# An ubiquitin-dependent balance between mitofusin turnover and fatty acids desaturation regulates mitochondrial fusion

Laetitia Cavellini[1,*], Julie Meurisse[1,*], Justin Findinier[1], Zoi Erpapazoglou[1], Naïma Belgareh-Touzé[1], Allan M. Weissman[2] & Mickael M. Cohen[1]

Mitochondrial integrity relies on homotypic fusion between adjacent outer membranes, which is mediated by large GTPases called mitofusins. The regulation of this process remains nonetheless elusive. Here, we report a crosstalk between the ubiquitin protease Ubp2 and the ubiquitin ligases Mdm30 and Rsp5 that modulates mitochondrial fusion. Ubp2 is an antagonist of Rsp5, which promotes synthesis of the fatty acids desaturase Ole1. We show that Ubp2 also counteracts Mdm30-mediated turnover of the yeast mitofusin Fzo1 and that Mdm30 targets Ubp2 for degradation thereby inducing Rsp5-mediated desaturation of fatty acids. Exogenous desaturated fatty acids inhibit Ubp2 degradation resulting in higher levels of Fzo1 and maintenance of efficient mitochondrial fusion. Our results demonstrate that the Mdm30-Ubp2-Rsp5 crosstalk regulates mitochondrial fusion by coordinating an intricate balance between Fzo1 turnover and the status of fatty acids saturation. This pathway may link outer membrane fusion to lipids homeostasis.

[1] Sorbonne Universités, UPMC University of Paris 06, CNRS, UMR8226, Laboratoire de Biologie Moléculaire et Cellulaire des Eucaryotes, Institut de Biologie Physico-Chimique, 75005 Paris, France. [2] Laboratory of Protein Dynamics and Signaling, Center for Cancer Research, NCI, Frederick, MD 21702, USA. * These authors contributed equally to this work. Correspondence and requests for materials should be addressed to M.M.C. (email: cohen@ibpc.fr).

Mitochondria assemble in a dynamic network whose plasticity is conditioned by fission and homotypic fusion of outer and inner membranes[1]. These membrane dynamics processes regulate all mitochondrial functions and are also essential for quality control within this organelle[2]. To allow for rapid response to the functional requirements of the mitochondrial network, the molecular machineries promoting fusion and fission of mitochondrial membranes must themselves be subject to exquisite regulation[1–3].

The ubiquitin–proteasome system (UPS) is a regulator of mitochondrial dynamics[4]. Ubiquitin, a 76 amino acid polypeptide, can covalently attach to specific lysine residues of target proteins[5]. Ubiquitin itself contains seven lysines to which additional ubiquitin moieties can be covalently linked resulting in a wide range of different poly-ubiquitin chains that promote different functions[6]. For example, K48-linked ubiquitin chains constitute a targeting signal for the 26S proteasome, while K63-linked chains mediate non-proteasomal regulatory functions. In conjunction with E2 ubiquitin-conjugating enzymes, E3 ubiquitin ligases provide the substrate and linkage specificity for ubiquitylation. Deubiquitylation enzymes (DUBs), of which ubiquitin specific proteases represent the largest family (UBPs in yeast) can, among other functions, trim ubiquitin chains and cleave ubiquitin moieties from substrates often antagonizing the action of specific E3s.

The impact of the UPS on mitochondrial dynamics is best exemplified by its action on mitofusins[7–14]. Mitofusins (MFN1/2 in mammals; Fzo1 in yeast) are large GTPases that belong to the superfamily of dynamin-related proteins. They constitute the core component of the fusion machinery for outer membranes[1,2]. Their auto-oligomerization in *cis* (that is, on the same outer membrane) and then in *trans* (that is, on outer membranes from opposing mitochondria) triggers tethering of mitochondria followed by homotypic fusion between outer lipid bilayers. Defects in this process are linked to neurodegenerative, metabolic and developmental disorders[1,15–17], but how fusion of outer membranes is regulated remains poorly understood.

In the yeast *Saccharomyces cerevisiae*, regulation of mitochondrial fusion involves the F-box protein Mdm30 (refs 7,18–21). Mdm30 is the substrate recognition element of the multisubunit ubiquitin ligase SCF^Mdm30, which promotes K48-ubiquitylation and subsequent degradation of Fzo1 (ref. 7). Overexpression of Fzo1 or its stabilization through Mdm30 inactivation, leads to inhibition of mitochondrial fusion[20,21]. This is manifested as a failure of cells to respire and major perturbations in the organization of mitochondrial networks. Fzo1 ubiquitylation and degradation are regulated by a DUB called Ubp2 (ref. 22). Ubp2 binds specifically to high molecular weight (MW) ubiquitylated species of Fzo1. Absence of the DUB not only induces accelerated turnover and low levels of Fzo1, but also causes defective mitochondrial fusion. Regulation of mitochondrial fusion by the UPS is thus governed by a paradox in which either down or upregulation of mitofusins decreases mitochondrial fusion efficiency. This paradox is not fully understood.

Ubp2 antagonizes ubiquitylation and degradation of Fzo1 promoted by an unknown E3 that has been proposed to be distinct from SCF^Mdm30 (ref. 22). In this regard, Ubp2 was originally identified as the antagonist of Rsp5 (refs 23,24), an ubiquitin ligase involved in a myriad of cellular functions in *S. cerevisiae*, especially the regulation of membranes homeostasis and trafficking events[25]. One essential function of Rsp5 is in the *OLE1*-pathway, where Rsp5 activates synthesis of the fatty acids desaturase Ole1 (refs 23,26–29), which is itself known to be critical for maintenance of mitochondrial homeostasis[30,31]. However, links between Rsp5 and Fzo1 have neither been investigated nor proposed.

Here, we show that Ubp2 antagonizes the turnover of Fzo1 that is mediated by Mdm30. We find that Ubp2 is also a substrate for Mdm30-dependent degradation, which in turn modulates Rsp5 function. This crosstalk between Mdm30, Ubp2 and Rsp5 is further demonstrated to maintain mitochondrial fusion efficiency not only through regulation of Fzo1 ubiquitylation but also through concomitant control of fatty acids desaturation. Our findings establish that regulation of fusion between mitochondrial outer membranes is conditioned by a balance between ubiquitylation of mitofusins and desaturation of fatty acids.

## Results

**Rsp5 is not involved in the regulation of Fzo1 by Ubp2.** Ubp2 antagonizes the ubiquitylation and degradation of Fzo1 promoted by an unknown E3 (ref. 22). Consistent with this, absence of Ubp2 resulted in lower endogenous levels of Fzo1 but did not alter expression of other factors involved in mitochondrial fission (Dnm1) or fusion (Ugo1, Mgm1) (Fig. 1a). To further evaluate Ubp2's specificity, the status of ubiquitylation of Fzo1 tagged at its C-terminus with 13Myc epitopes (Fzo1-13Myc) was monitored in mutants of *S. cerevisiae* each lacking expression of 1 of 16 UBP DUBs (Fig. 1b and Supplementary Fig. 1a). After normalizing the loading of whole cell extracts to Fzo1-13Myc levels (Fig. 1b; anti-Myc short expo), the level of the characteristic Mdm30-dependent doublet of ubiquitylated Fzo1[7] was not substantially altered in any of the retested UBP mutants. However, a smear of higher MW forms migrating above the ubiquitylation doublet was specifically observed in the *ubp2Δ* mutant (Fig. 1b; anti-Myc long expo). These observations confirm the specific involvement of Ubp2 in the regulation of Fzo1 ubiquitylation.

Given Ubp2 is an antagonist of Rsp5 (refs 23,24), we tested the involvement of this E3 in the regulation of Fzo1. Neither ubiquitylation (Supplementary Fig. 1b) nor degradation (Supplementary Fig. 1c) of Fzo1-13Myc were affected by deletion of *RSP5* (kept viable with the Spt23 p90 fragment which rescues the essential *OLE1*-Pathway in *rsp5Δ* cells, Supplementary Table 1 and Fig. 4d). Most importantly, the Fzo1-13Myc smear seen with *UBP2* deletion persisted in *rsp5Δ* + *spt23** *ubp2Δ* cells (Fig. 1c; compare lane 3 with lane 4). Similarly, the faster degradation of endogenous Fzo1 (Fig. 1d) or of Fzo1-13Myc expressed at physiological levels (Supplementary Fig. 1d) in *UBP2* null cells was unaffected by the absence of Rsp5. These results indicate that Rsp5 is not implicated in the regulation of Fzo1 by Ubp2.

Rup1 is a binding factor of Ubp2 and Rsp5 that promotes the interaction of the DUB with the E3 and regulates the Ubp2-dependent reversal of Rsp5-mediated ubiquitylation specifically[23,32–34]. However, deletion of *RUP1* did neither impact the steady state levels of Fzo1 (Fig. 1e) nor its high MW species (Fig. 1f). This confirms that Rsp5 is not involved in the regulation of Fzo1 ubiquitylation and degradation by Ubp2.

**Ubp2 antagonizes Mdm30-mediated degradation of Fzo1.** Absence of Ubp2 induces accumulation of ubiquitin K48 linkages on Fzo1 and accelerates its turnover[22]. Ubp2 may thus antagonize Mdm30, which promotes K48-ubiquitylation and degradation of Fzo1 (ref. 7). Assessment of Fzo1-13Myc turnover by cycloheximide (CHX) chase, confirmed that lack of Ubp2 accelerates Fzo1 degradation and revealed that absence of Mdm30 stabilizes the mitofusin whether or not *UBP2* is also deleted (Fig. 2a and Supplementary Fig. 2a). This specific involvement of Ubp2 and Mdm30 in the regulation of Fzo1 degradation was further confirmed by assessing the turnover of endogenous Fzo1 as compared to other proteins (Supplementary Fig. 2b). These results

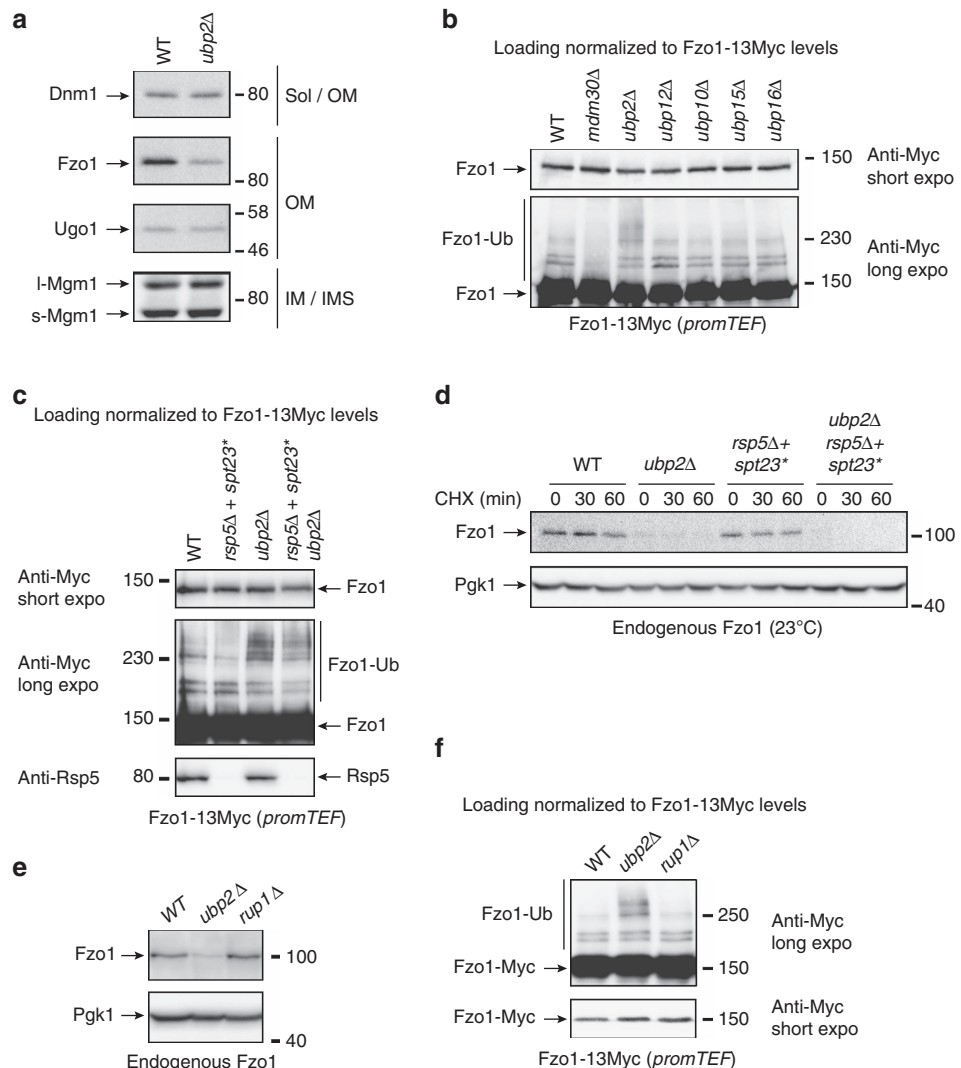

**Figure 1 | The impact of Ubp2 on Fzo1 is not mediated by Rsp5. (a–f)** MW in kDa are shown on the right or left of short and long exposures of indicated regions of immunoblots. (**a**) Total protein extracts prepared from WT and *ubp2Δ* cells (W303 background) were analysed by anti-Dnm1, anti-Fzo1, anti-Ugo1 and anti-Mgm1 immunoblotting. The localization of proteins is indicated on the right. (**b**) Total protein extracts prepared from WT, *mdm30Δ* and *ubpXΔ* cells (BY4741 background) transformed with pRS416-TEF-FZO1-13MYC were analysed by anti-Myc immunoblotting. Loading was normalized to Fzo1-13Myc levels. (**c**) Total protein extracts prepared from WT, *rsp5 + spt23\**, *ubp2Δ* and *rsp5Δ + spt23\* ubp2Δ* cells (DF5 background) transformed with pRS414-TEF-FZO1-13MYC were analysed by anti-Myc and anti-Rsp5 immunoblotting. Loading was normalized to Fzo1-13Myc levels. (**d**) CHX chase analysed by anti-Fzo1 and anti-Pgk1 immunoblotting with strains used in **c** but free of any plasmid. (**e**) Total protein extracts prepared from WT, *ubp2Δ* and *rup1Δ* cells (BY4741 background) were analysed by anti-Fzo1 and anti-Pgk1 immunoblotting. (**f**) Total protein extracts prepared from cells used in **e** but transformed with pRS414-TEF-FZO1-13MYC were analysed by anti-Myc immunoblotting. Loading was normalized to Fzo1-13Myc levels. Sol, soluble; OM, outer membrane; IM, inner membrane; IMS, inter membrane space; l-Mgm1, long Mgm1; s-Mgm1, short Mgm1.

indicate that Mdm30 is essential for the accelerated turnover of Fzo1 that is induced by the absence of Ubp2.

Consistent with this, both the higher MW smear of Fzo1-13Myc seen in *ubp2Δ* cells as well as the characteristic Mdm30-dependent ubiquitylation doublet were absent in *ubp2Δ mdm30Δ* mutants (Fig. 2b and Supplementary Fig. 2c). Therefore, Mdm30 is also essential for the generation of all Fzo1 higher MW species we managed detecting and that were previously shown to bind specifically to Ubp2 (ref. 22).

Out of the 78 lysine residues of Fzo1, the Mdm30-dependent ubiquitylation doublet is conjugated to lysine 398 of the mitofusin[22]. Mutation of lysine 398 into arginine (K398R) not only prevented formation of the doublet in WT cells but also blocked generation of all Fzo1 higher MW species seen in absence of Ubp2 (Fig. 2c). Since *UBP2* deletion increases ubiquitin K48

linkages on Fzo1 (ref. 22), this result indicates that Ubp2 restricts K48-elongation of the Mdm30-dependent ubiquitin doublet.

However, Ubp2 targets disassembly of K63-linked chains predominantly[23,24]. Therefore, we monitored the *in vitro* capacity of mock ( − Ubp2) and HA ( + Ubp2) immunoprecipitates from *UBP2-HA* cells to trim K63 or K48-linked chains composed of 5 ubiquitin moieties. These reactions were carried out in the presence of DTT or $H_2O_2$ as the K63-activity of Ubp2 is activated by reduction but totally inhibited by oxidation[35].

As expected, Ubp2 cleaved one to three ubiquitin moieties from Ub5K63 chains, which was stimulated by increasing concentrations of DTT but totally abrogated by $H_2O_2$ (Fig. 2d; left panels). The activity of Ubp2 on Ub5K48 chains was also sensitive to the redox environment but restricted to the removal of a single ubiquitin molecule without detectable decrease in

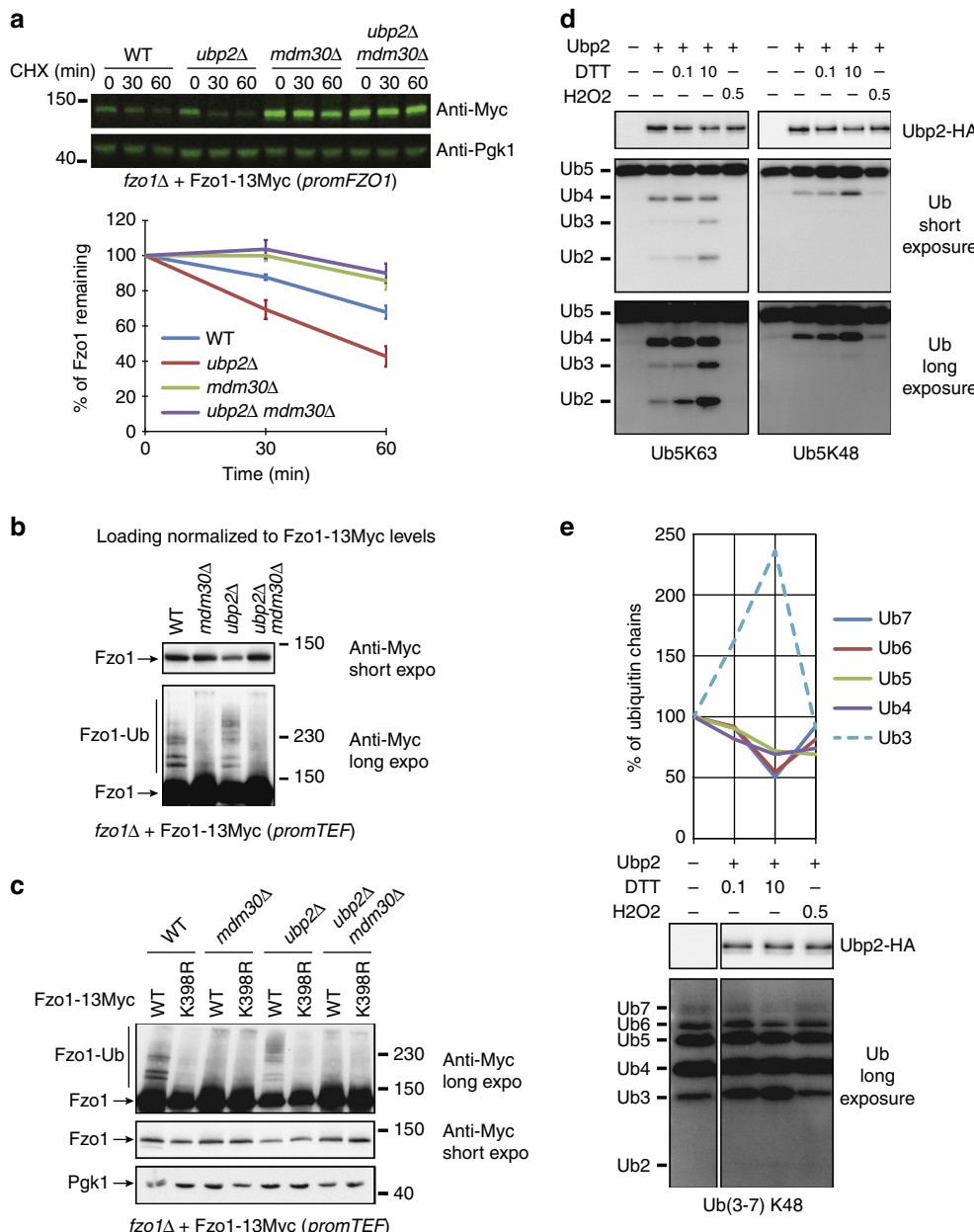

**Figure 2 | Ubp2 antagonizes the degradation of Fzo1 that is mediated by Mdm30.** (**a**) Quantification of CHX chases (bottom) with *WT*, *mdm30Δ*, *ubp2Δ* and *ubp2Δ mdm30Δ* cells (MCY572 background) shuffled with pRS414-FZO1-13MYC were analysed by anti-Myc and anti-Pgk1 immunoblotting followed by detection with fluorescent secondary antibodies (top). Error bars represent s.e.m. from three independent experiments. (**b,c**) MW in kDa are shown on the right of short and long exposures of indicated regions of anti-Myc or anti-Pgk1 immunoblots. (**b**) Total protein extracts prepared from strains used in **a** but shuffled with pRS414-TEF-FZO1-13MYC were analysed by anti-Myc immunoblotting. Loading was normalized to Fzo1-13Myc levels. (**c**) Total protein extracts prepared from *FZO1*, *mdm30Δ*, *ubp2Δ* and *ubp2Δ mdm30Δ* cells (MCY572 background) shuffled with pRS414-TEF-FZO1-13MYC (WT) or pRS414-TEF-FZO1-13MYC K398R were analysed by anti-Myc and anti-Pgk1 immunoblotting. (**d**) *In vitro* deubiquitylation assays. Ub5K63 or Ub5K48 chains were incubated with mock (−) or Ubp2-HA immunoprecipitates in the absence (−) or in the presence of DTT (0.1 or 10 mM) or H2O2 (0.5 mM). Reactions were analysed by anti-HA and anti-Ub immunoblotting. (**e**) Bottom: same as (**d**) but with Ub(3-7) K48 chains. Top: quantification of the level of each length of chain relative to the mock (−Ubp2) condition.

Ub5K48 substrate, even in the presence of 10 mM DTT (Fig. 2d; right panels). Ubp2 can thus use K48-linked chains of 5 ubiquitin moeities as substrates with low efficiency but is unable to process chains of 4 ubiquitins. This capacity of Ubp2 to disassemble long but not short K48-linked ubiquitin chains was further assessed with chains composed of 3 (Ub3) to 7 (Ub7) ubiquitin moieties. In the presence of increasing concentrations of DTT, all types of K63-linked chains efficiently diminished to generate Ub2 species

(Supplementary Fig. 2d). In contrast, Ubp2 decreased the amounts of K48-linked Ub6 and Ub7 chains and concomitantly increased those of K48-linked-Ub3 chains but had limited effect on K48-linked Ub5 and Ub4 chains (Fig. 2e). Ubp2 thus cleaves Ub5K48 chains with low efficiency to generate Ub4K48 (Fig. 2d; right panels) and Ub6 or Ub7K48 chains with higher efficiency to generate Ub3K48 (Fig. 2e). All Ubp2 effects on K63 and K48-linked chains were inhibited with H2O2. These results

confirm that while Ubp2 can trim K63-linked chains independent of their size, its ability to disassemble K48 linkages is restricted to longer chains of more than 4–5 Ub moieties. Importantly, this likely explains the stability of the characteristic Fzo1 ubiquitylation doublet *in vivo*.

Thus, while Ubp2 had been reported to antagonize the UPS-mediated degradation of Fzo1 mediated by an unknown, non-Mdm30-dependent ligase[22], the data presented herein suggest that this DUB restricts the extension of Mdm30-dependent K48 ubiquitin chains, which limits the turnover of Fzo1. While it is still unclear whether Mdm30 directly mediates the addition of K48 linkages beyond the Fzo1 ubiquitin doublet, our results establish that Ubp2, by limiting this extension, is able to antagonize the Mdm30-mediated degradation of Fzo1.

**Ubp2 is a substrate for Mdm30-mediated degradation.** Lack of Ubp2 induces a defect in mitochondrial fusion presumably caused by lower levels of Fzo1 (ref. 22). This defect is manifest as a specific respiratory deficiency of *ubp2Δ* cells at 37 °C, as assessed by growth on selective media containing either dextrose (fermentable) or glycerol (non-fermentable) as the sole carbon source (Supplementary Fig. 3a).

Since we identified Mdm30 as an E3 essential for promoting the faster turnover of the mitofusin (Fig. 2), we employed a shuffling strain strategy (Methods section) to test whether stabilization of Fzo1 through deletion of *MDM30* could correct the respiratory growth in cells lacking Ubp2. However, at 37 °C, absence of *MDM30* annihilated glycerol growth and had a dominant effect over deletion of *UBP2* (Fig. 3a; compare *mdm30Δ* and *mdm30Δ ubp2Δ*). Most surprisingly, absence of the DUB led to a marked rescue in respiratory growth of *mdm30Δ* cells at both 23 and 30 °C (Fig. 3b and Supplementary Fig. 3b; compare *mdm30Δ* and *mdm30Δ ubp2Δ*).

To begin addressing the molecular basis of this finding, we monitored expression of Ubp2-HA in *mdm30Δ* cells that express either WT Mdm30-Myc or a non-functional version, in which the F-box motif of Mdm30-Myc is mutated, making it unable to bind to other SCF components and form a functional ubiquitin ligase. Expression of this inactive form induced a two-fold increase in levels of Ubp2-HA (Fig. 3c), raising the possibility that Ubp2 might be subject to Mdm30-dependent proteasomal degradation. Consistent with this, levels of Ubp2-HA increased upon treatment with the proteasome inhibitor MG132 (Fig. 3c). Moreover, the DUB bound to Mdm30 in co-immunoprecipitation experiments (Fig. 3d). The level of WT Mdm30-Myc was much lower than that of the F-box mutant (Fig. 3d; lysates) because F-box proteins are themselves targeted for proteasomal degradation when assembled as part of an SCF ubiquitin ligase complex[7,36,37]. Ubp2-HA was co-immunoprecipitated (IP) with the F-box mutant but not with the wild-type version of Mdm30-Myc (compare IP WT with IP f-box). The interaction between Ubp2 and Mdm30 is thus stabilized when Mdm30 is not part of the SCF. This may be explained by the relative abundance of Mdm30-Myc and Ubp2-HA in WT and F-box mutant conditions. Alternatively, Ubp2 may be rapidly degraded following binding to WT Mdm30 rendering this interaction too transient to be detected in WT conditions.

Genomic Ubp2 tagged with six HA C-terminal epitopes (Ubp2-HA) increased in *mdm30Δ* as compared to *WT* cells, whether CCCP, a potent uncoupler of oxidative phosphorylation, was added or not (Fig. 3e). This indicates that increased Ubp2 levels do not result from general mitochondrial dysfunction but from the absence of Mdm30. Notably, growth in respiratory conditions also stimulated the production of high MW forms of Ubp2, which are Mdm30-dependent (Fig. 3e; red arrow),

suggesting Ubp2 ubiquitylation by the Mdm30 ubiquitin ligase. To determine whether Mdm30 controls Ubp2 turnover, the degradation of genomically tagged Ubp2-HA was monitored by CHX chase in *WT* and *mdm30Δ* cells (Fig. 3f and Supplementary Fig. 3c). Consistent with Mdm30-dependent turnover, Ubp2 was degraded with a half-life of greater than 60 min which was significantly increased in *mdm30Δ* cells. The levels of genomically tagged Ubp2-HA did not increase upon deletion of *RUP1* (Supplementary Fig. 3d) and Ubp2-HA was co-IPed with the F-box mutant of Mdm30-Myc regardless of the presence or the absence of Rup1 (Supplementary Fig. 3e). Therefore, Rup1 is not involved in the Mdm30-dependent degradation of Ubp2.

The finding that Mdm30 controls the level of Ubp2 provides an explanation for the rescue seen in *mdm30Δ ubp2Δ* cells (Fig. 3b and Supplementary Fig. 3b). Mitochondrial defects resulting from absence of Mdm30 would be induced by accumulation of Ubp2 but abrogated upon deletion of the DUB. Consistent with this, *mdm30Δ* cells overexpressing Ubp2 grew markedly slower on glycerol media than mutants transformed with a control vector at both 23 and 30 °C (Fig. 3g and Supplementary Fig. 3f). This demonstrates that the respiratory deficiency in *mdm30Δ* cells is due not only to stabilization of Fzo1 but also involves accumulation of Ubp2.

**Stabilized Ubp2 downregulates Rsp5 functions in *mdm30Δ* cells.** Accumulation of Ubp2 plays a causal role in respiratory defects seen in *mdm30Δ* cells (Fig. 3). This suggests that stabilized Ubp2 antagonizes the ubiquitylation of one or several substrates which alters mitochondrial function in cells that lack Mdm30. Consequently, increasing ubiquitylation of these Ubp2 targets or increasing the level of the E3 antagonized by Ubp2, should phenocopy the absence of Ubp2 (Fig. 3b) and positively impact respiration in *mdm30Δ* strains. Consistent with this, excess Rsp5 (Fig. 4a) resulted in partial restoration of respiratory growth in *mdm30Δ* cells (Fig. 4b and Supplementary Fig. 4a). Since *MDM30* deletion does not alter the levels or stability of endogenous Rsp5 (Supplementary Fig. 4b), these results suggest that accumulation of Ubp2 induces downregulation of Rsp5-mediated functions, which contributes to respiratory defects in *mdm30Δ* cells.

This prompted us to assess mitochondrial fusion in *rsp5Δ + spt23** cells using an *in vivo* mitochondrial fusion assay. During yeast mating, mitochondria from each haploid cell, differentially labelled with mito-GFP (green fluorescent protein) and mito-RFP (red fluorescent protein), fuse together. The extent of co-localization between these fluorophores serves as readout for evaluating the efficiency of mitochondrial fusion *in vivo*. Total co-localization between fluorescent proteins was observed in all zygotes obtained from mating between haploid *rsp5Δ + spt23** strains, which were kept viable through ectopic activation of the *OLE1*-pathway (Fig. 4c). This indicates that if the *OLE1*-pathway is maintained, all other Rsp5-dependent functions can be lost without impacting mitochondrial fusion.

In the *OLE1*-pathway (Fig. 4d), Rsp5 activates the synthesis of the Δ9-fatty acid desaturase Ole1. This enzyme catalyses the desaturation of the carbon 9–carbon 10 bonds of palmitic (16:0) and stearic acids (18:0) to produce palmitoleic (16:1) and oleic (18:1) acids, which are essential for cell survival and maintenance of mitochondrial distribution[31]. Importantly, endogenous levels of both the Ole1 protein (Fig. 4e; *OLE1-9MYC* genomically tagged) and *OLE1* transcripts (Fig. 4f and Supplementary Fig. 4c) were decreased in *mdm30Δ* cells but were restored by either excess of Rsp5 or absence of Ubp2. Stabilization of Ubp2 in *mdm30Δ* cells thus results in downregulation of the *OLE1*-pathway.

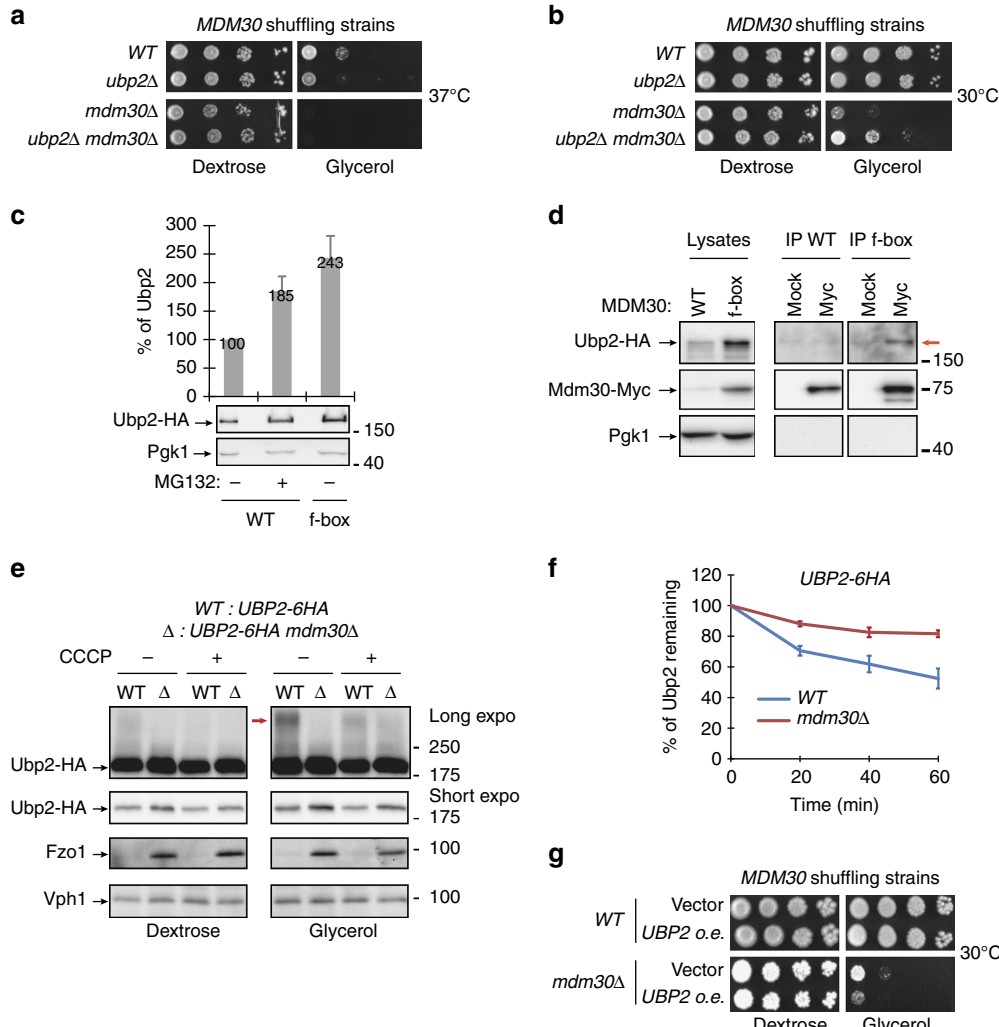

**Figure 3 | Ubp2 is a degradation substrate of Mdm30.** (**a,b**) Dextrose and glycerol spot assays at 37 °C (**a**) or 30 °C (**b**) of *MDM30* (MCY971) and *MDM30 ubp2Δ* (MCY996) shuffling strains before (*WT* and *ubp2Δ*) or after (*mdm30Δ* and *mdm30Δ ubp2Δ*) cure of the *MDM30* shuffle plasmid. (**c–e**) MW in kDa are shown on the left of indicated regions of immunoblots. (**c**) Total protein extracts prepared from *ubp2Δ mdm30Δ* cells co-expressing Ubp2-HA and WT or f-box mutant Mdm30-Myc, treated ( + ) or untreated ( − ) with MG132, were analysed by anti-Myc and anti-Pgk1 immunoblotting (bottom) and quantified relative to WT without MG132 (Top). Error bars represent the s.d. from three independent experiments. (**d**) Strains from (**c**) were lysed and lysates were subjected to co-IP with anti-Myc or mock antibody followed by immunoblotting as indicated. Left panels, lysates (10% input of IP); right panels, immunoprecipitates. Red arrow indicates co-IP between Ubp2-HA and the f-box mutant of Mdm30-Myc. Pgk1 was used as a loading and IP control. (**e**) Total protein extracts from *UBP2-6HA* (MCY968) and *UBP2-6HA mdm30Δ* (MCY1031) strains grown for 90 min in YPD (Dextrose) or YPG (Glycerol) in the presence ( + ) or in the absence ( − ) of 1 mM CCCP were analysed by anti-HA, anti-Fzo1 and anti-Vph1 immunoblotting. The Mdm30-dependent high MW species of Ubp2-HA are indicated (red arrow). (**f**) Quantification of CHX chases with *UBP2-6HA* (MCY968) and *UBP2-6HA mdm30Δ* (MCY1031) strains analysed by anti-HA and anti-Pgk1 immunoblotting followed by detection with fluorescent secondary antibodies. Error bars represent the s.e.m. from three independent experiments. (**g**) Dextrose and glycerol spot assays of *MDM30* shuffling strains (MCY971) transformed with pRS423-UBP2-6HA (*UBP2 o.e.*) or an empty vector, before (*WT*) or after (*mdm30Δ*) cure of the *MDM30* shuffle plasmid.

**Increased UFAs results in stabilization of Ubp2 and Fzo1.** A 38% decrease in the Ole1 expression induces drastic accumulation of lipids with saturated acyl chains resulting in extensive remodelling of the intracellular phospholipidome[38]. This may explain the importance of Ole1 in maintenance of mitochondrial distribution[31] as phospholipids and their derivatives actively participate in outer membranes fusion[39,40]. In this context, the 25% Ole1 decrease in *mdm30Δ* cells (Fig. 4e) may have similar effects on phospholipids and the imbalance between low levels of Ole1 and high levels of Fzo1 might be causal in promoting the mitochondrial fusion deficiency in cells that lack Mdm30.

Mitochondrial fusion in *WT* cells was not affected by exogenous oleic acid (Fig. 5a), one of the desaturation reaction products of Ole1 and the most abundant unsaturated fatty acid (UFA) in *S. cerevisiae*. This indicates that increased desaturation of fatty acids is compatible with efficient mitochondrial fusion *in vivo*. Remarkably, however, UFAs not only induced increased levels of endogenous Fzo1 but also promoted accumulation of genomically tagged Ubp2-HA in both respiratory and fermentative growth conditions (Fig. 5b). The impact of UFAs on Ubp2 and Fzo1 levels was abolished in *mdm30Δ* cells (Fig. 5b). The increased level of Fzo1 in WT cells could be due either to stabilization of Ubp2 (*cf* Fig. 3) or loss of Mdm30 activity towards Fzo1. The latter possibility was excluded as treatment of *ubp2Δ* cells with UFAs did not significantly affect Mdm30 levels (Supplementary Fig. 5a) and did not decrease

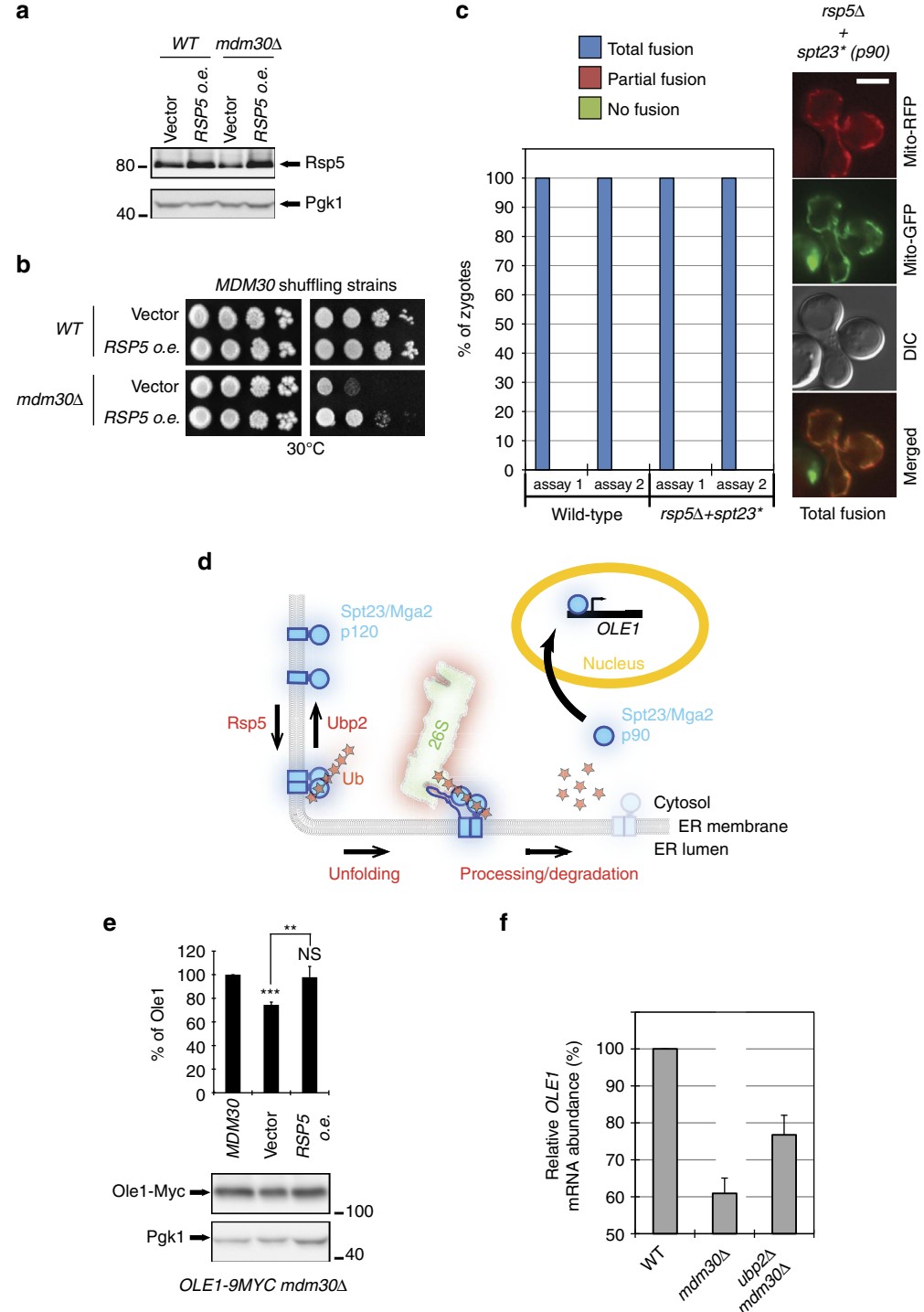

**Figure 4 | Rsp5 functions are down-regulated in *mdm30Δ* cells.** (**a**) Total protein extracts from *MDM30* shuffling strains (MCY971) transformed with pRS315-RSP5 or an empty vector and covered by (*WT*) or cured from (*mdm30Δ*) the *MDM30* shuffle plasmid were analysed by anti-Rsp5 and anti-Pgk1 immunoblotting. (**b**) Dextrose and glycerol spot assays with strains from (**a**). (**c**) *In vivo* mitochondrial fusion assays with WT (MCY408 and MCY415 strains) or *rsp5Δ + spt23\** (MCY881 and MCY882 strains) haploid cells of opposing mating types and expressing mito-RFP or mito-GFP. All zygotes obtained from mating (50 in each assay), displayed total co-localization (total fusion) between mito-GFP and mito-RFP (example shown on the right; scale bar 5 μm). (**d**) Schematic description of the *OLE1*-pathway (see text). In the *OLE1*-pathway, two ER-resident transmembrane proteins, Spt23 and Mga2, homodimerize when the saturation of lipids acyl chains increases. This triggers Rsp5-mediated ubiquitylation of the full length proteins (p120 forms), which promotes their proteasomal endoproteolysis, releasing soluble N-terminal fragments (p90 forms) that function as transcription factors for the synthesis of the Δ9-fatty acid desaturase Ole1. (**e**) Bottom: Total protein extracts from *OLE1-9MYC mdm30Δ* strains (MCYO2 parental strain) containing *MDM30*, *RSP5* or empty plasmids were analysed by anti-Myc and anti-Pgk1 immunoblotting; Top: Quantification of Ole1-Myc levels in *mdm30Δ* relative to *MDM30* cells. Error bars represent the s.d. from three independent experiments. \*\**P* < 0.05, \*\*\**P* < 0.005 (one-way analysis of variance (ANOVA)). NS, not significant. (**f**) Relative *OLE1* mRNA levels in *WT*, *mdm30Δ* and *ubp2Δ mdm30Δ* strains (MCY971 parental strain) determined by qPCR. Error bars represent the s.d. from six reactions.

either the degradation (Supplementary Fig. 5b) or ubiquitylation (Fig. 5c) of Fzo1. In fact, an ~50% increase in the Mdm30-dependent ubiquitylation doublet, consistent with the increase in Ubp2 levels, was observed in WT cells treated with UFAs

(Fig. 5c). In *ubp2Δ* cells, this doublet decreased by 50% at the benefit of Fzo1 high MW species regardless of the addition of oleic acid. These findings indicate that WT cells naturally adapt to increased UFAs by inhibiting degradation of Ubp2, thus resulting

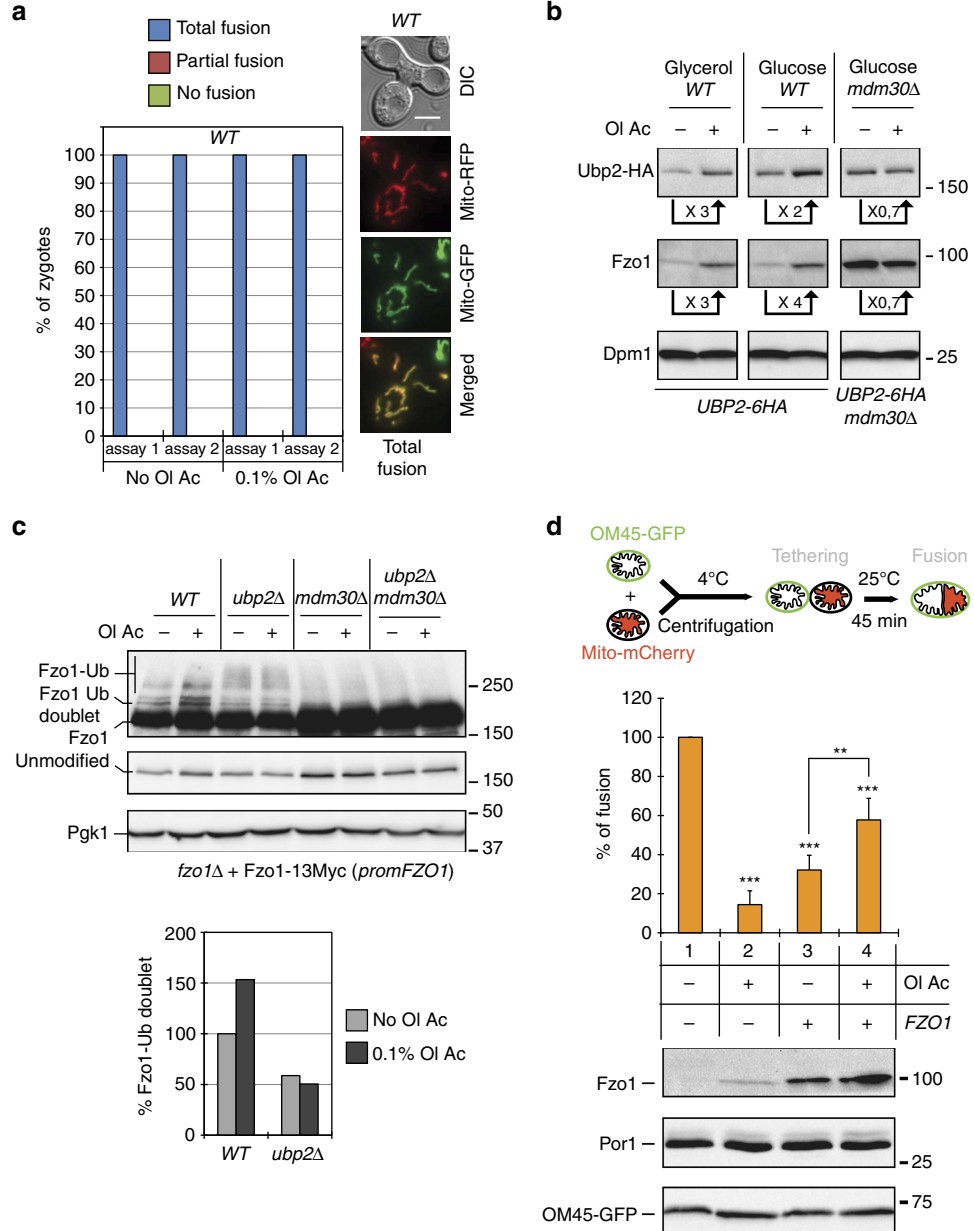

**Figure 5 | Excess UFAs induce natural increase in Ubp2 and Fzo1 levels.** (**a**) *In vivo* mitochondrial fusion assays with *WT* haploid cells of opposing mating types (MCY970 and MCY971 strains) with or without 0.1% oleic acid. In two independent experiments (assay 1 and assay 2), all zygotes obtained from mating (38 in each assay), displayed total co-localization (total fusion) between GFP and RFP (example shown on the right; scale bar 5 μM). (**b,c**) MW in kDa are shown on the right or left of immunoblots. (**b**) Total protein extracts from *UBP2-6HA* (MCY968) and *UBP2-6HA mdm30Δ* (MCY1031) strains grown in YPG (glycerol) or YPD (glucose) with ( + ) or without ( − ) 0.1% oleic acid were analysed by anti-HA, anti-Fzo1 and anti-Dpm1 immunoblotting. Quantification of fold increase ( > 1) or decrease ( < 1) of Ubp2-HA and Fzo1 normalized to Dpm1 in treated ( + Ol Ac) relative to untreated ( − Ol Ac) conditions is indicated. (**c**) Top: Total protein extracts prepared from *WT*, *mdm30Δ*, *ubp2Δ* and *ubp2Δ mdm30Δ* cells (MCY572 background) shuffled with pRS414-FZO1-13MYC and grown in the absence ( − ) or in the presence ( + ) of 0.1% oleic acid were analysed by anti-Myc and anti-Pgk1 immunoblotting. Fzo1 high MW species (Fzo1-Ub), the Fzo1-Ub doublet and unmodified Fzo1 are indicated on the left of immunoblots. Bottom: quantification of the Fzo1-Ub doublet normalized to Pgk1 levels and relative to the untreated (-Ol Ac) WT condition. (**d**) Top: *in vitro* outer membrane fusion assays with mitochondria from OM45-GFP and Mito-mCherry cells. Center: reactions were performed with mitochondria purified from cells transformed with pRS314-FZO1 ( + pFZO1) or an empty vector ( − pFZO1) and either treated ( + Ol Ac) or untreated ( − Ol Ac) with 0.1% oleic acid. The level of outer membrane fusion (6%) was set at 100% in untreated conditions (reaction 1). % of outer membrane fusion efficiency in reactions 2, 3 and 4 was calculated relative to reaction 1. Error bars represent the s.d. from three independent experiments. \*\*P < 0.05, \*\*\*P < 0.005 (ANOVA). More than 1,000 mitochondria were measured per sample. Bottom: protein extracts from reactions were analysed by anti-Fzo1, anti-GFP and anti-Por1 immunoblotting.

in inhibition of ubiquitin chain elongation and stabilization of Fzo1.

To extend this observation, we assessed the effects of UFAs on mitochondrial fusion *in vitro*[41]. Mitochondria were purified from cells in which the outer membrane protein OM45 was genomically tagged with GFP or in which genomically integrated mCherry was fused to a mitochondrial matrix-targeting signal. These mitochondria were mixed and stage 1 fusion reactions, which allow for outer but not inner membranes fusion[19,41], were carried out (Fig. 5d). Outer membrane fusion was assessed by GFP and mCherry signals co-localization (Supplementary Fig. 6a), and this approach was validated by inhibition of co-localization with agents that block mitochondrial fusion[41,42] (Supplementary Fig. 6b).

When outer membrane fusion reactions were carried out using mitochondria purified from cells grown in 0.1% oleic acid (+Ol Ac), *in vitro* fusion was decreased by 86% (Fig. 5d graph; compare column 2 to 1). This is despite upregulation of Fzo1 in response to UFAs was preserved in the *in vitro* preparations (Fig. 5d blots). This result contrasts with the lack of effect of UFAs on *in vivo* fusion reactions (Fig. 5a). *In vivo*, the impact of UFAs on Fzo1 levels may translate in the accumulation of Fzo1 molecules at the tip of mitochondrial tubules undergoing fusion. Since mitochondrial tubules undergo extensive fragmentation during mitochondrial isolation, the concentration of Fzo1 molecules may not be homogenous over the whole population of purified mitochondria, unless Fzo1 levels further increase. Consistent with this, an extra copy of *FZO1* (Fig. 5d blots) decreased *in vitro* fusion by 68% in the absence of UFAs (Fig. 5d graph; compare column 3 to 1) but induced significant rescue of outer membrane fusion efficiency in the presence of oleic acid (Fig. 5d graph; compare column 4 to 2). These results reveal that the increase in intracellular UFAs directly affects the fusogenicity of outer membranes *in vitro* and that increased levels of Fzo1 molecules can overcome this defect.

**Increased UFAs and Fzo1 improve fusion in *mdm30Δ* cells.** A concomitant increase in UFAs and of Fzo1 expression maintains *in vivo* mitochondrial fusion (Fig. 5a,b) and stimulates fusion of outer membranes *in vitro* (Fig. 5d). In this context, the imbalance between low levels of Ole1 and high levels of Fzo1 may contribute to promoting the mitochondrial fusion deficiency in cells that lack Mdm30.

To test this, *MDM30* null haploid strains of opposing mating types harbouring a control vector or an *MDM30* plasmid and expressing either mito-GFP or mito-RFP were grown with or without 0.1% oleic acid before mating. On the basis of the extent of co-localization between GFP and RFP in resulting zygotes, three categories of mitochondrial networks were identified (Fig. 6a): those with perfect co-localization over the whole zygote (total fusion); those with co-localization in zygote daughter cells only (partial fusion); those with lack of co-localization over the whole zygote (no fusion).

Total fusion occurred in 98% of *MDM30* zygotes whereas partial fusion prevailed in 90% of zygotes lacking Mdm30 (Fig. 6b), which is consistent with the Mdm30 requirement for efficient mitochondrial fusion[21]. Addition of 0.1% oleic acid induced a limited rescue of total fusion in *mdm30Δ* zygotes transformed with the control vector (Fig. 6c; vector). Treating *mdm30Δ* cells with 0.1% oleic acid may have pushed the imbalance between Fzo1 and UFAs the other way around with levels of Fzo1 remaining high but limiting and desaturation of fatty acids becoming stronger.

Fusion assays were thus repeated with *mdm30Δ* cells transformed with an *FZO1* expression plasmid (Supplementary Fig. 7).

Strikingly, total fusion in zygotes containing the extra *FZO1* copy increased from 20% in untreated, to 40% in conditions treated with UFAs (Fig. 6c). The mitochondrial fusion defect in *mdm30Δ* cells can therefore be corrected by concomitant increase in fatty acids desaturation and Fzo1 expression.

**Functional mitochondrial fusion upon decreased Fzo1 and UFAs.** Our data establish that increased levels of UFAs diminish the efficiency of mitochondrial outer membranes fusion, a defect that is prevented by accumulation of mitofusins. This suggests that mitochondrial fusion may be regulated by a balance between Fzo1 degradation and the status of fatty acids saturation/desaturation. Validation of this hypothesis would require that decreased desaturation of fatty acids combined with increased Fzo1 loss would also support efficient mitochondrial fusion.

To test this, *UBP2* null cells are ideal as Fzo1 levels are decreased due to enhanced degradation (Fig. 7a). This is because loss of Ubp2 expression results in unopposed Mdm30-mediated turnover (Fig. 2), which is accompanied by a mitochondrial fusion defect[22]. To test whether this defect can be rescued by decreased UFAs, *MGA2* was deleted. This resulted in a greater than 80% decrease in *OLE1* mRNAs in the *UBP2 MGA2* double mutant as compared to WT cells (Supplementary Fig. 8a). This strong downregulation of the *OLE1*-pathway induced a partial rescue in the defect of respiratory growth (Fig. 7b and Supplementary Fig. 8b), but did not modify the mitochondrial morphology changes (Fig. 7c) caused by the absence of Ubp2. These results suggest that mitochondrial fusion is only partially restored when decreased Fzo1 levels get combined with decreased UFAs.

Notably, deletion of *MGA2*, which results in decreased UFAs, affects neither respiratory growth (Fig. 7b) nor mitochondrial morphology (Fig. 7c), indicating that mitochondrial fusion is fully functional in *mga2Δ* cells. Moreover, deletion of *MGA2* does not impact Fzo1 levels (Fig. 7a). These findings appear to argue against a balance between Fzo1 loss and decreased UFAs. Strikingly, however, when the turnover of endogenous Fzo1 was monitored in *mga2Δ* cells, Fzo1 degradation was found to be accelerated compared to WT cells (Fig. 8a). As the steady state level of Fzo1 is not affected in cells lacking Mga2 (Fig. 7a), it is likely that Fzo1 biosynthesis is upregulated in these cells to compensate for more rapid degradation. Most importantly, this suggests that it indeed may be an increased rate of Fzo1 degradation that is key to promoting efficient mitochondrial fusion when UFAs are decreased. If this is the case, then further increasing the amount of Fzo1 degraded in *ubp2Δ mga2Δ* cells should enhance the partial restoration of function seen in Fig. 7.

To accomplish this, we introduced an extra copy of WT *FZO1* in *ubp2Δ* and *ubp2Δ mga2Δ* mutants. Remarkably, this *FZO1* extra copy induced total rescue of glycerol growth (Fig. 8b, Supplementary Fig. 9a,b) and restoration of tubular mitochondrial morphology (Fig. 8c) in *ubp2Δ mga2Δ* cells. However, it did not rescue the defects in respiration and morphology in the *ubp2Δ* single mutant (Fig. 8b,c; *ubp2Δ* cells) where levels of Ole1 do not decrease significantly (Supplementary Fig. 8a and ref. 43). These marked differences in function and fusion between *ubp2Δ* and *ubp2Δ mga2Δ* cells were observed despite the levels of mitofusins being similarly low whether or not the *FZO1* extra copy was expressed (Fig. 8d). The positive effect induced by additional Fzo1 in *ubp2Δ mga2Δ* cells is therefore very likely caused by an enhanced rate of Fzo1 degradation, not an increase in the steady state levels of the mitofusin. Consistent with this, the total rescue of glycerol growth in *ubp2Δ mga2Δ* cells at 37 °C was fully dependent on K398 of Fzo1 (Fig. 8e and Supplementary

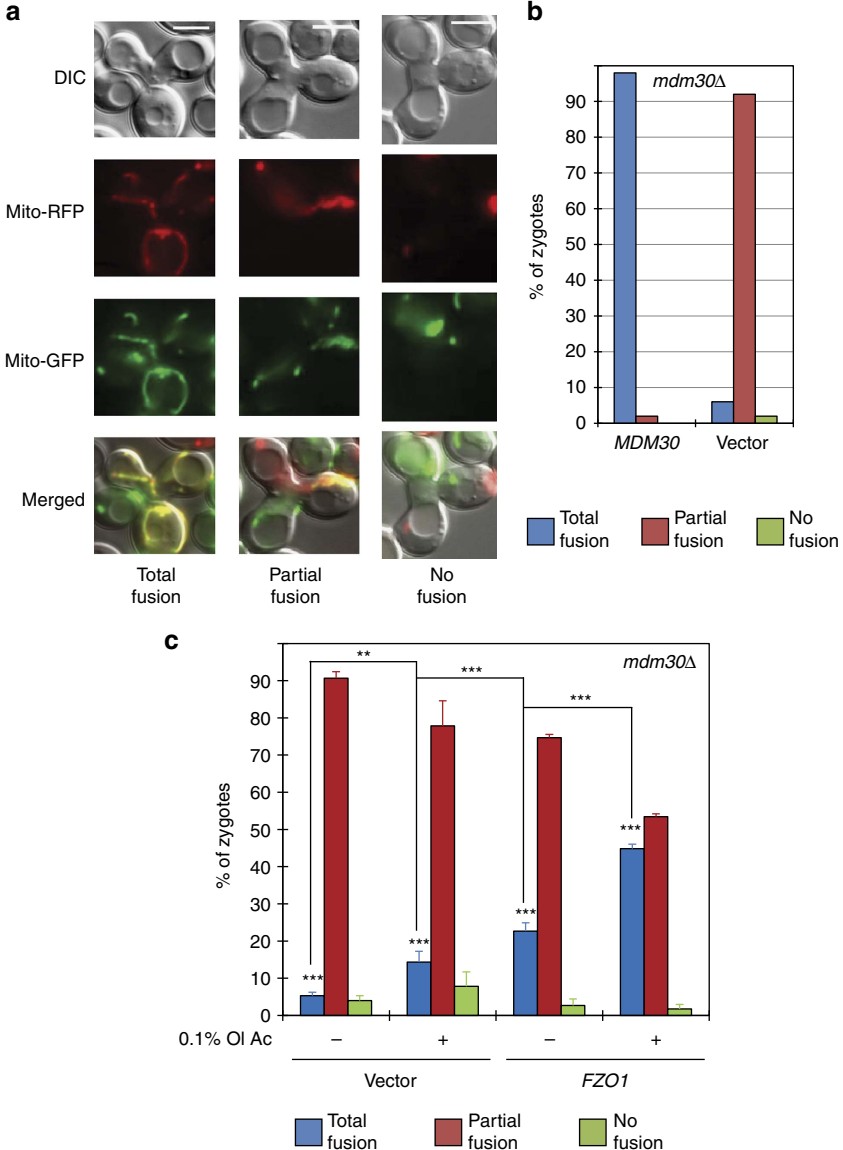

**Figure 6 | Increase in UFAs and Fzo1 rescue mitochondrial fusion in *mdm30Δ* cells. (a–c)** *In vivo* mitochondrial fusion assays after mating between *mdm30Δ* haploid cells of opposing mating types (MCY970 and MCY971 parental strains) containing *MDM30*, *FZO1* or empty plasmids and expressing either mito-RFP or mito-GFP, respectively. (**a**) Examples of zygotes with mitochondrial networks totally fused, partially fused or not fused (scale bar, 5 μM). (**b,c**) Percentage of indicated zygotes with total (blue), partial (red) or no (green) mitochondrial fusion obtained from indicated cells grown in the absence or in the presence of 0.1% oleic acid. Error bars in **c** represent the s.d. from three independent experiments. **$P<0.05$, ***$P<0.005$ (one-way analysis of variance (ANOVA)). More than 60 zygotes were analysed per sample.

Fig. 9c), which is essential for formation of the prominent Mdm30-dependent ubiquitylation doublet.

These results demonstrate that Mdm30-dependent ubiquitylation and degradation of Fzo1 plays an active role in mitochondrial fusion upon decreased fatty acids desaturation. Importantly, they establish that mitochondrial fusion is regulated by a balance between the rate of degradation of Fzo1 and the status of fatty acids saturation/desaturation.

## Discussion
Ubp2 has been proposed to antagonize the degradation of Fzo1 mediated by an unknown E3 that is distinct from Mdm30 (ref. 22). In this previous study, high MW species of Fzo1 were found to accumulate in *ubp2Δ* cells that express a catalytic mutant of this DUB. These species were not only shown to interact specifically with the catalytic mutant of Ubp2 but the vast majority also disappeared in the absence of Mdm30, which would be consistent with a potential function of the DUB as an antagonist of Mdm30. Yet, because of residual high MW species persisting in *ubp2Δ mdm30Δ* cells, it was concluded that Ubp2 does not antagonize Mdm30-mediated ubiquitylation[22]. Here we confirm that absence of Ubp2 results in stabilization of high MW species of Fzo1 and accelerated degradation of the mitofusin. By analysing these features in *ubp2Δ mdm30Δ* cells, we determine that absence of Mdm30 abolishes all the effects of Ubp2 on Fzo1 (Fig. 2). While we do not exclude that other E3s may play a role in controlling Fzo1 levels, our results indicate that Ubp2 antagonizes the Mdm30-mediated turnover of Fzo1.

We find that besides acting as an antagonist, Ubp2 is also a substrate for Mdm30-dependent degradation (Fig. 3). This revealed that respiratory defects in *mdm30Δ* cells are not strictly

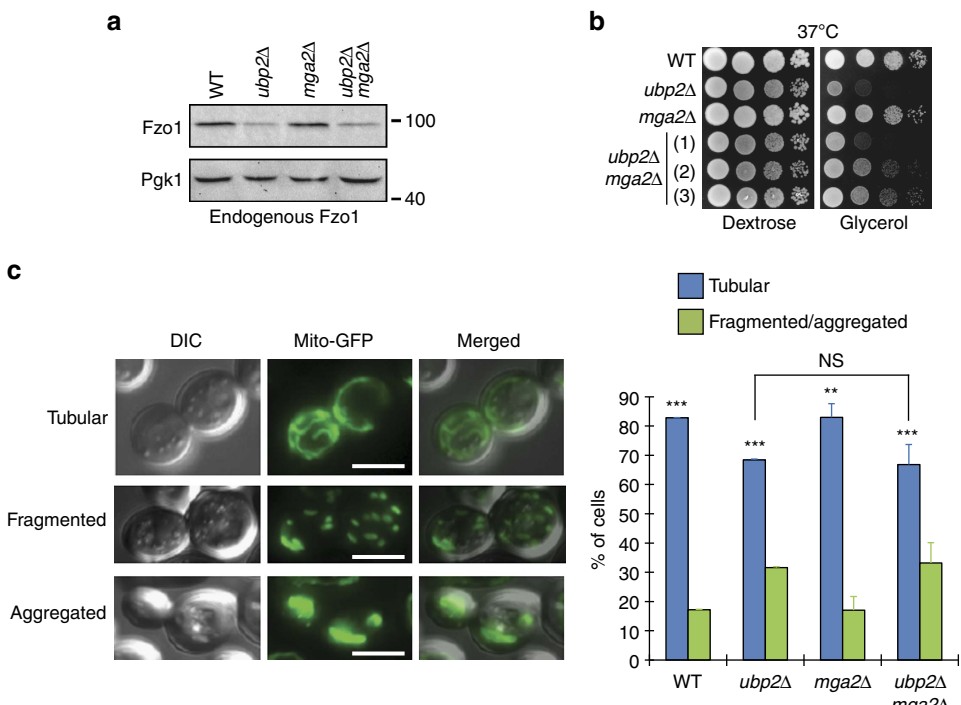

**Figure 7 | Low Fzo1 levels and low UFAs support only partial mitochondrial fusion. (a)** Total protein extracts prepared from *WT*, *ubp2Δ*, *mga2Δ* and *ubp2Δ mga2Δ* cells (DF5 background) were analysed by anti-Fzo1 and anti-Pgk1 immunoblotting. MW in kDa are shown on the right of indicated regions of immunoblots. **(b)** Dextrose and glycerol spot assays with strains from **(a)**. Three distinct clones of *ubp2Δ mga2Δ* cells were analysed. **(c)** Left: examples of cells expressing mitochondrial matrix-targeted GFP (mito-GFP) with tubular, fragmented or aggregated mitochondrial networks are shown; Scale bar, 5 μM. Right: Percentage of cells with tubular (blue) or fragmented/aggregated (green) mitochondria from strains used in **a**. Error bars represent the s.d. from three independent experiments. **\*\*P<0.05, \*\*\*P<0.005** (one-way analysis of variance (ANOVA)). NS, not significant. More than 95 cells were analysed per sample.

caused by accumulation of mitofusins but also involve stabilization of Ubp2. As Ubp2 and Rsp5 are functional antagonists, this stabilization translates into downregulation of Rsp5-dependent functions as exemplified by attenuation of the *OLE1*-pathway in cells that lack Mdm30 (Fig. 4). Consequently we unravel a crosstalk between mitofusin turnover and fatty acids desaturation in the regulation of mitochondrial fusion. Increased levels of UFAs results in decreased efficiency of outer membrane fusion, which is overcome by Fzo1 stabilization (Figs 5 and 6). Conversely, in the setting of decreased synthesis of UFAs, defects in mitochondrial fusion can be overcome by increased degradation of Fzo1 (Figs 7 and 8). Mitochondrial fusion efficiency therefore depends on coordination of a balance between the levels of Fzo1 and the status of fatty acids saturation (Fig. 9). We not only provide strong evidence that this balance naturally takes place in WT cells but also that it is implemented through Mdm30-mediated control of Fzo1 and Ubp2 levels (Fig. 9, red square). Exogenous UFAs induce inhibition of Mdm30-mediated degradation of Ubp2, which antagonizes degradation of Fzo1 and results in increased levels of the mitofusin (Fig. 5). These observations position the regulation of Ubp2 by Mdm30 as a potential lever in the Fzo1 degradation/fatty acids desaturation balance. It will be of interest to dissect how this regulation is modulated.

We demonstrate that Mdm30-mediated ubiquitylation and degradation of Fzo1 allows for tuning the levels of mitofusins according to the status of fatty acids desaturation. This agrees with the integral role of Fzo1 degradation in the fusion process[19] and resolves an apparent paradox where either high or low levels of mitofusins result in defective mitochondrial fusion. In parallel, our results raise the question of how fatty acids saturation might impact mitochondrial lipid bilayers. Fatty acids are precursors of lipids, and their saturation status directly affects the composition of intracellular phospholipids[38]. In turn, phospholipids and their derivatives participate in fusion of mitochondrial outer membranes in conjunction with mitofusins[39,40]. Excess UFAs in WT cells results in diminished capacity of outer membranes to fuse *in vitro* and this defect is rescued by increased levels of mitofusins (Fig. 5d). Stabilization of Fzo1 may thus favour fusion of outer membranes rendered less 'fusogenic' upon increased desaturation of fatty acids. The phospholipids composition of outer membranes that result from decreased desaturation of fatty acids could, on the other hand, compensate for decreased levels of mitofusin. Alternatively, it may be that the relative amount of UFAs, by altering the saturation status of the lipid bilayer, may modulate the accumulation of Fzo1 molecules at sites of mitochondrial contact. Future investigations will require a dissection of the potential impact of fatty acids saturation on the phospholipid composition of mitochondrial membranes as well as a more in-depth understanding of the organization of mitofusins at the mitochondrial surface.

Some crosstalk between mitofusins and fatty acids have been observed in mammalian cells, particularly in systems dealing with study of mechanisms leading to insulin resistance, diabetes and obesity[44–46]. Our findings potentially open new avenues of investigation regarding these and other metabolic disorders.

## Methods
**Yeast strains and growth conditions.** The *S. cerevisiae* strains and plasmids used in this study are listed in Supplementary Tables 1 and 2, respectively. Standard methods were used for growth, transformation and genetic manipulation of *S. cerevisiae*. Minimal synthetic media [Difco yeast nitrogen base (Voigt Global

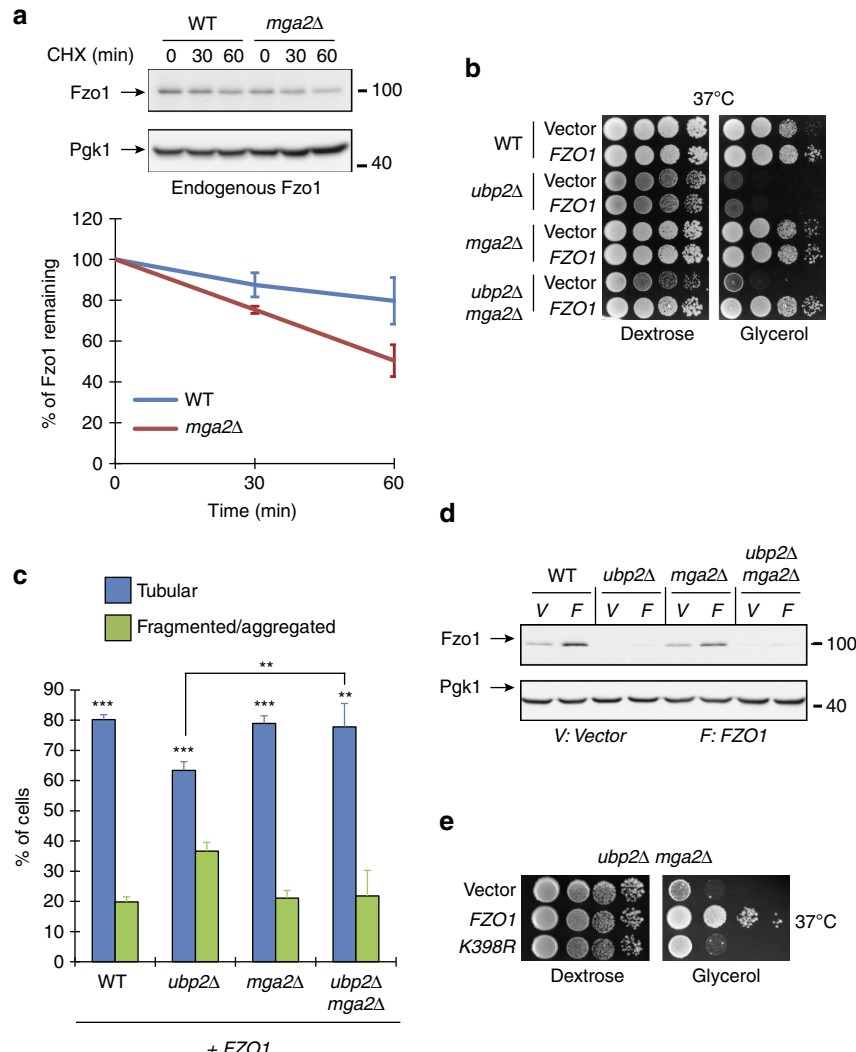

**Figure 8 | Fast Fzo1 turnover is compatible with mitochondrial fusion upon low UFAs. (a)** Top: CHX chase analysed by anti-Fzo1 and anti-Pgk1 immunoblotting with *WT* and *mga2Δ* cells (DF5 background). MW in kDa are shown on the right of indicated regions of immunoblots. Bottom: quantification of CHX chases. Error bars represent the s.e.m. from three independent experiments. **(b)** Dextrose and glycerol spot assays with *WT*, *ubp2Δ*, *mga2Δ* and *ubp2Δ mga2Δ* (clone 3) cells (DF5 background) transformed with an empty plasmid (Vector) or pRS314-FZO1 (*FZO1*). **(c)** Percentage of cells with tubular (blue) or fragmented/aggregated (green) mitochondria from strains used in **b** but transformed with pRS314-FZO1. Error bars represent the s.d. from three independent experiments. **P < 0.05, ***P < 0.005 (one-way analysis of variance (ANOVA)). NS, not significant. More than 95 cells were analysed per sample. **(d)** Total protein extracts prepared from strains in **b** were analysed by anti-Fzo1 and anti-Pgk1 immunoblotting. MW in kDa are shown on the right of indicated regions of immunoblots. **(e)** Dextrose and glycerol spot assays with *ubp2Δ mga2Δ* (clone 3) cells (DF5 background) transformed with an empty plasmid (Vector), pRS314-FZO1 (*FZO1*) or pRS314-FZO1 K398R (*K398R*).

Distribution Inc, Lawrence, KS), and drop out solution] supplemented with 2% dextrose (SD; YPD for complete media), 2% glycerol (SG; YPG for complete media) or 2% raffinose (SR) were prepared as described[47]. In the indicated strains (see Supplementary Table 1), *FZO1, MDM30, UBP2 and OLE1* were chromosomally deleted or C-terminally tagged using conventional homologous recombination approaches[48,49]. Treatment with MG132 (BostonBiochem) was performed at 150 μM in the presence of Proline and SDS in the growth media[50]. Oleic acid (Sigma-Aldrich) was added to media at indicated concentrations in the presence of 1% Tergitol (Sigma-Aldrich). Where indicated, cells were treated with 1 mM CCCP (Sigma-Aldrich) for 90 min. The primers for generation of plasmids and strains used in this study are listed in Supplementary Tables 3 and 4.

**Generation of FZO1 and MDM30 shuffle strains.** Similarly to *FZO1* null cells, strains lacking *MDM30* tend to lose their mitochondrial DNA because of decreased mitochondrial fusion efficiency[21]. Consequently, reliable genetic analysis of *mdm30Δ* cells warranted resorting to a plasmid-shuffle strategy. To generate *FZO1* and *MDM30* shuffle strains, *MATa* and *MATα* W303 cells (MCY553 and MCY554) were mated together yielding a diploid strain in which *FZO1* or *MDM30* were chromosomally deleted. Resulting strains heterozygous for *FZO1* and *MDM30* were transformed with pRS416-FZO1 and pRS316-MDM30 (*FZO1* and *MDM30*

shuffle plasmids), respectively. Following sporulation in 1% potassium acetate for 7 days at 24 °C, tetrad dissections were performed on a SINGER MSM400 device. *fzo1Δ* and *mdm30Δ* spores covered by *FZO1* and *MDM30* shuffling plasmids were selected on appropriate media.

To yield strains used in the study but not listed in Supplementary Table 1, *FZO1* and *MDM30* shuffle strains were transformed with indicated plasmids under selection of interest. Ten colonies were systematically isolated on SD selective media lacking uracil and replica-plated on corresponding SG or 5′-fluoroorotic acid (5′-FOA) plates. Strains grown on 5′-FOA plates and cured from shuffling plasmids were in turn replica-plated on SD and SG selective plates containing uracil. The glycerol growth phenotypes of strains covered by or cured from the shuffling plasmids were reproducibly observed in 100% of clones tested after 1 to 3 days of growth at 30 °C. Representative colonies were used in subsequent experiments.

**Protein extracts and immunoblotting.** Cells grown in SD were collected during the exponential growth phase (OD$_{600}$ = 0.5–1). Total protein extracts were prepared by the NaOH/trichloroacetic acid lysis technique[51]. Proteins were separated by SDS–PAGE 8% and transferred to nitrocellulose membranes (Amersham Hybond-ECL; GE Healthcare). The primary antibodies used for

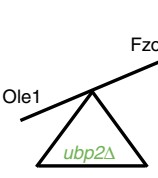
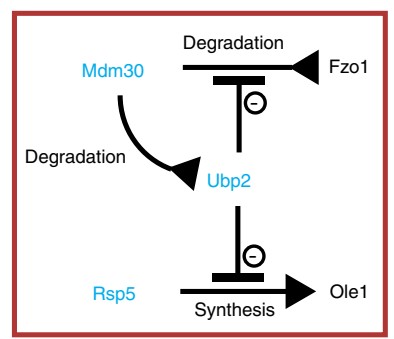
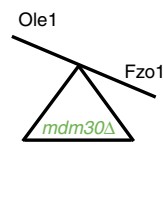

**Figure 9 | Model of mitochondrial fusion regulation by the UPS.** Functional mitochondrial fusion requires a maintained balance between Fzo1 and Ole1 levels in WT cells (Red). In *ubp2Δ* and *mdm30Δ* mutants, this balance is affected in opposite directions resulting in deficient mitochondrial fusion (Green). The central red square depicts the enzymes (Mdm30, Ubp2 and Rsp5) and their crosstalks that allow maintaining the Ole1/Fzo1 balance. As indicated, synthesis, degradation and inhibition are symbolized by normal, inverted and flat arrowheads, respectively.

immunoblotting were monoclonal anti-Pgk1 (1/20,000, AbCam, ab113687), monoclonal anti-HA (1/1,000, 12CA5, Invitrogen, 71-5500), monoclonal anti-Por1 (1/1,000, AbCam, ab110326), monoclonal anti-Dpm1 (1/1,000, AbCam, ab113686), monoclonal anti-Vph1 (1/1,000, AbCam, ab113683), monoclonal anti-Ub (1/1,000, P4D1, Santa Cruz Biotechnology, sc-8017), monoclonal anti-myc (1/1,000, 9E10, Invitrogen, R950-25), monoclonal anti-GFP (1/1,000, Roche, 11814460001), polyclonal anti-Fzo1 (1/1,000, generated by Covalab), polyclonal anti-Ugo1 (1/1,000, generated by Covalab), polyclonal anti-Dnm1 (1/1,000, generated by Covalab), polyclonal anti-Mgm1 (1/1,000, Kind gift from Andreas Reichert) and polyclonal anti-Rsp5 (1/5,000, Kind gift from Linda Hicke). Primary antibodies were detected by secondary anti-mouse or anti-rabbit antibodies conjugated to horseradish peroxidase (HRP, 1/5,000, Sigma-Aldrich, 12-348 and A5278), followed by incubation with the Immun-Star Western C Kit (Bio-Rad). Immunoblotting images were acquired with a Gel Doc XR + (Bio-Rad) and treated with the Image Lab 3.0.1 software (Bio-Rad). Uncropped scans of the most important blots of the study are shown in Supplementary Fig. 10.

**Anti-ubiquitin immunoblotting.** Following separation by SDS–PAGE, proteins were transferred to PVDF (polyvinylidene difluoride) membranes (immobillion-FL, Merck Millipore), activated by incubation in 100% ethanol for 5 min at RT. After transfer, membranes were boiled in water for 15 min, before blocking (TBST + 5% BSA) and incubation with a monoclonal anti-ubiquitin antibody coupled with HRP (P4D1-HRP conjugate, 1/3,000, Enzo, BML-PW0935) diluted in TBST + 5% BSA, for 1 h at room temperature, followed by three washes of 10 min each in TBST before revelation as described above.

**Cycloheximide chase and quantification.** To monitor turnover of proteins, CHX (Sigma-Aldrich) was added to yeast cultures growing at 37 °C (unless indicated) at a final concentration of $100 \, \mu g \, ml^{-1}$. Total protein extracts were prepared at the indicated time points after addition of CHX. For quantification purposes, fluorescent secondary antibodies (1/5,000, IRDye 800 anti-mouse; LiCor, 926-32210) were used and proteins detected and quantified using an Odyssey Fc scanner (LiCor) and Image Studio 2.0 analysis software (LiCor). Levels of Fzo1-13Myc and Ubp2-6HA were then normalized to those of Pgk1 for each time point and calculated relative to those of time 0 from each condition. Data reported are the s.e.m. (error bars) from three independent experiments.

**Co-immunoprecipitations.** For co-immunoprecipitation assays between Mdm30 and Ubp2, the *MDM30 ubp2Δ* shuffle strain (MCY996) was cured from the *MDM30* shuffle plasmid (pRS316-MDM30) and transformed with either pTEF-MDM30-MYC or pTEF-fbox-MYC. Resulting transformants were subsequently transfected with pRS423-UBP2-6HA to yield strains that co-express HA and MYC tagged versions of Ubp2 and Mdm30 as the sole source of both proteins. These cells grown at exponential phase were lysed at 4 °C with glass beads in IP buffer (100 mM Tris, pH 7.5, 100 mM sodium chloride, 0.6% Triton X-100, 10% glycerol, supplemented with protease inhibitors (Protease Inhibitor Cocktail; Sigma-Aldrich) and 1 mM Pefabloc (Sigma-Aldrich)). Insoluble material was removed by centrifugation for 30 min at 13,000g. Aliquots of supernatants were diluted and heated in sample buffer (pre-IP lysate). The remaining supernatant was

incubated for 2 h at 4 °C with protein G-Sepharose beads (rec-Protein G-Sepharose 4B Conjugate, Invitrogen) in the absence (Mock) or in the presence of anti-Myc antibodies (9E10, Invitrogen, R950-25). Beads washed with IP buffer were heated in sample buffer before resolution by SDS–PAGE and analysis by immunoblotting with indicated antibodies.

***In vitro* deubiquitylation assays.** For *in vitro* deubiquitylation assays, immuno-precipitations with Protein G magnetic beads (Pierce) and anti-HA (12CA5, Invitrogen, 71-5500) were performed using lysates obtained from *ubp2Δ mdm30Δ* cells that co-express C-terminally epitope-tagged versions of Ubp2 (Ubp2-HA) and Mdm30 (Mdm30-Myc). Beads were subsequently subjected to serial dilutions and incubated for 1 h at 25 °C with 0.5 µg Ub5K63 or Ub5K48 chains or Ub(3-7)K63 or Ub(3-7)K48 chains (BostonBiochem) in 50 µl reaction buffer (10 mM Tris, pH 7.5, 50 mM sodium chloride, 5 mM magnesium chloride) supplemented with indicated concentrations of DTT or $H_2O_2$. Reactions were stopped by heating samples in sample buffer.

**Spot assays.** Cultures grown overnight in SD medium were pelleted, resuspended at $OD_{600} = 1$, and serially diluted (1:10) five times in water. Three microliters of the dilutions were spotted on SD and SG plates and grown for 2 to 4 days (Dextrose) or 3 to 6 days (Glycerol) at 23, 30 or 37 °C.

**Quantification of Ole1 mRNA levels.** 2 ODs of cells grown in SD media to exponential phase ($OD_{600} = 0.5$–1) were spheroplasted by treatment with 2.5 U per OD of Zymolyase (Zymo Research; Orange, CA). The total RNA was extracted using the NucleoSpin RNA extraction kit (Macherey Nagel). 1 µg of resulting RNAs were then treated with DNase I (NEB) and subjected to reverse transcriptionusing Maxima First Strand cDNA synthesis Kit for quantitative PCR with reverse transcription (Thermo Scientific). Quantitative real time PCR was performed with Maxima SYBR Green qPCR Master Mix (Thermo Scientific) in a Bio-Rad CFX96 Real-Time PCR system using *OLE1* and *ACT1* specific primers. The *OLE1* mRNA abundance in mutants was normalized to *ACT1* mRNA levels and expressed relative to *OLE1* expression in WT using the Livak method. Six experiments were performed and the relative mRNA levels were averaged.

**Quantification of Ole1 protein levels.** *In vivo* expression of Ole1 was analysed using *MDM30* shuffle strains in which the chromosomal copy of *OLE1* is tagged with 9Myc (MCYO2) or 13Myc (MCY1126 and MCY1032) epitopes. *OLE1-9MYC* cells covered by (*MDM30*) or cured from (*mdm30Δ*) the *MDM30* shuffle plasmid were grown in SD medium overnight and processed for protein extracts once reaching the exponential phase ($OD_{600} = 0.5$–1). Following anti-Myc and anti-Pgk1 immunoblotting, levels of Ole1-Myc and Pgk1 were quantified by densitometry with the Image Lab 3.0.1 software (Bio-Rad). After normalization to levels of Pgk1, amounts of Ole1-Myc in all strains were calculated relative to those of *MDM30* positive cells. Data reported in graphs are the mean and s.d. from three independent experiments.

**Mitochondrial network morphology.** Mitochondrial morphology was scored in cells expressing mito-GFP from pYX232-mtGFP plasmids. Strains were grown in dextrose medium to mid-log phase and fixed with 3.7% formaldehyde. Morphology phenotypes were assessed in at least 100 cells. Data reported are the mean and s.d. (error bars) from three independent experiments.

**In vivo fusion assays.** Mitochondrial fusion in zygotes was examined essentially as described[52]. Indicated *MATa* and *MATα* were respectively transformed with pYeL1-mtGFP and pYeL1-mtRFP that allow expression of mito-GFP or mito-RFP under control of the GAL10 promoter. For each fusion assay, haploid cells of opposing mating types were grown overnight in SR media complemented with 2% Galactose in the presence or in the absence of 0.1% oleic acid. Next morning, expression of mtGFP and mtRFP were repressed for 2 h by addition of 2% Glucose in cultures. *MATa* and *MATα* cells were then mixed in SD media and mated during 4 h in the presence or in the absence of 0.1% oleic acid before analysis by fluorescence microscopy. Mitochondrial fusion was quantified in 38 to 70 large-budded zygotes per strain in three separate experiments. Data reported are the mean and s.d. of all experiments.

**In vitro mitochondrial fusion assays.** Mitochondrial fractions for *in vitro* fusion assays[53] were prepared as follows. Cells were grown to stationary phase in dextrose medium before being cultured in glycerol medium to reach a final $OD_{600}$ of 0.8–1.0. For cell walls disruption, cells were first incubated in β-mercaptoethanol buffer (100 mM Tris-HCl pH 9.4; 50 mM β-mercaptoethanol) for 20 min at 30 °C and then in 1.2 M sorbitol supplemented with zymolyase (Zymo Research; USA) for 30 min. Spheroplasts were dounced 100 times in cold NMIB buffer (0.6 M sorbitol, 5 mM $MgCl_2$, 50 mM KCl, 100 mM KOAc, 20 mM Hepes pH 7.4), before centrifugation of the lysate at 3,000 × g for 5 min at 4 °C. The resulting supernatant was centrifuged at 10,170g for 10 min at 4 °C and protein concentration in the mitochondrial enriched pellet was determined with the Bradford assay (Bio-Rad Protein Assay; Bio-Rad Laboratories GmbH, Germany).

Fusion reactions were carried out as follows. OM45-GFP mitochondria of 0.25 mg was mixed with 0.25 mg of mito-mCherry mitochondria (mixing step) before centrifugation at 10,170g for 10 min at 4 °C. After incubation of mitochondrial pellets for 10 min on ice to allow mitochondrial docking, the supernatant was replaced by Stage 1 buffer (20 mM Pipes–KOH pH 6.8, 150 mM KOAc, 5 mM Mg(OAc)₂, 0.6 M sorbitol) and fusion of outer membranes was induced by incubation at 25 °C for 45 min. Fusion reactions were stopped by fixation with two volumes of 8% formaldehyde in phosphate-buffered saline. Aliquots were immobilized on microscope slides by mixing 1:1 with 2% low melting point agarose (Sigma-Aldrich) in NMIB.

The ratios of fused mitochondria were obtained as follows. The amount of GFP and mCherry signals co-localizing with each other were divided with the total number ($n > 1,000$) of OM45-GFP and mito-mCherry mitochondria (obtained from strains #779 and #980, respectively) in both reactions stopped at the mixing step and reactions stopped at $t = 45$ min. Fusion ratios at the mixing step were subtracted from fusion rations at $t = 45$ min to reveal the levels of fused mitochondria in each condition.

**Fluorescence microscopy.** Fluorescence microscopy was carried out with a Zeiss Axio Observer.Z1 microscope (Carl Zeiss S.A.S.) with a × 63 oil immersion objective equipped with the following filter sets: FITC (Filter set 44, Excitation BP 475/40, Beam Splitter FT 500, Emission BP 530/50) for GFP, Propidium Iodide (Filter set 00, Excitation BP 530–585, Beam Splitter FT 600, Emission LP 615) for RFP. Cell contours were visualized with Nomarski optics. Images were acquired with an ORCA-R2 charge-coupled device camera (Hamamatsu). Images were treated and analysed with ImageJ.

**Data availability.** The authors declare that all the data supporting the findings of this work are available within the article and its Supplementary Information files and available from the corresponding author upon reasonable request.

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

## Acknowledgements

We thank C. Lepage, C. Muther and M. Raymond for excellent technical assistance within UMR8226 and Zhou Xu for critical reading of the manuscript. Research in the Cohen laboratory is supported by the CNRS-INSERM ATIP-Avenir program, the 'fondation pour la recherche médicale' (INE 20100518343), a Marie Curie IRG grant (No. 276912) to M.M.C and the labex DYNAMO (ANR-11-LABX-0011-DYNAMO). Research in the Weissman laboratory is supported by the National Institutes of Health, National Cancer Institute, Center for Cancer Research, USA.

## Author contributions

L.C., J.M. and J.F. performed the experiments. L.C., J.M. and M.M.C. analysed the data. M.M.C. and N.B.T. designed the experiments. Z.E. and A.M.W. participated in the initiation of the project. M.M.C. conceived the project and wrote the manuscript with contributions of all the authors.

## Additional information

**Competing interests:** The authors declare no competing financial interests.

