## [Peer Review File · Nature Communications]

Reviewers' Comments:

Reviewer #1 (Remarks to the Author)

The manuscript by Cavellini, et al., describes relationships between Ubp2 (a deubiquitinating enzyme), Fzo1 (a mitofusin), and two ubiquitin ligases (Mdm30 and Rsp5). Many of the central conclusions are poorly substantiated and there is a lack of molecular understanding of several of the key observations. Given this, the contribution of the work to the field appears to be minimal. Also, the manuscript is in need of extensive editing. The meaning of several key statements is nearly completely lost due to poor sentence structure and word choice.

Specific comments:

It is unclear why the authors use the word "canonical" when referring to Ubp2 as a "canonical antagonist" of Rsp5. While Ubp2 is an antagonist of Rsp5, their relationship/interaction/mechanism does not follow any general rule and is therefore not canonical of anything.

Figure 1: After several readings, I still have no understanding of what is meant by Fzo1 "short" and "long".

Page 7, first paragraph: an example of the confusion in the writing. It is first said that the "absence of Mdm30 stabilized the mitofusin whether or not UBP2 was also deleted", then two sentences later it is stated that "these results indicate that the accelerated turnover of Fzo1 that is induced by the absence of Ubp2 is abolished in the absence of Mdm30." The first statement is simple and direct, while the second implies some type of regulatory "order" to the system that simply does not exist in either the experiment or in nature.

The section on K63 versus K48 chain type specificity of Ubp2 is confusing. The in vitro results are consistent with Ubp2 having very low activity against K48 chains (and perhaps no activity against K48 chains in cells, but this is difficult to know). However, the results are interpreted as being consistent with the fact that Mdm30 catalyzes K48 chain formation on Fzo1. This makes seem to make no sense. The types of chains formed on Fzo1 by Mdm30 must be addressed by determining the type of polyubiquitin chains that are accumulating on Fzo1 in the ubp2 deletion mutant - either by mass spec of the purified protein or perhaps by analysis in a K63R yeast strain. Also, while it is shown that all ubiquitylation of Fzo1 is dependent on K398, I don't see how this bears on the question of chain type. The general issue here is that if Mdm30 is directly ubiquitylating Fzo1 and Ubp2 is directly reversing this reaction, then Mdm30 and Ubp2 are expected to be synthesizing and removing, respectively, the same type of chains.

The section entitled "Ubp2 is a physiological substrate for Mdm30-mediated degradation" is underwhelming and incomplete. The changes in Ubp2 levels appear marginal, as does the co-IP of Mdm30 and Ubp2. Also, the types of chains that are accumulating on Ubp2 should again be examined. Are these really K48 chains, as would be expected if catalyzed by Mdm30, or are they K63 chains (perhaps catalyzed by its association with Rsp5). It is unclear whether attempts were made at in vitro reconstitution of the interactions between Mdm30 and Ubp2 and Fzo1, but this is what is clearly needed to get any molecular insight into what is going on here.

Page 11, lines 8-12: The conclusions made in the last two sentences of this paragraph are vague and confusing. "Convey" is a poor word choice - should this be replaced by suggest/imply, or demonstrates/indicates? The last sentence of the paragraph is worse, as I am unclear what the evidence is that Mdm30 affects the myriad activities of Rsp5.

Figure 4c (mitochondrial fusion assay): It is entirely unclear why the authors found it surprising that there was total co-localization of fluorophores in the *rsp5*-delta strain. There are absolutely no controls for this experiment - just one highly engineered experimental strain was examined (*rsp5*-delta expressing Spt23 "p90").

Reviewer #2 (Remarks to the Author)

Herein, Cavellini and colleagues describe the antagonistic effects of the deubiquitylase enzyme Ubp2 and of the ubiquitin ligase Mdm30 on the mitofusin Fzo1 in yeast and on mitochondrial function. While Ubp2 deletion destabilizes Fzo1, Mdm30 deletion stabilizes it. They then describe an interesting cross talk between the Ole1 pathway of lipid desaturation (in which Ubp2 plays a role) and mitochondrial fission, indicating that Fzo1 levels must be tightly regulated in the face of fluctuations in lipid saturation.

The study is original, unexpected and interesting.

The data are in general of excellent quality and support the most important conclusions of the study. It is a bit unfortunate that the authors did not go as far as testing that the various treatments indeed affect mitochondrial lipid composition as expected, but this is not crucial at this point.

Some additional work and some rewriting could nevertheless improve the manuscript.

Here are the points that need to be addressed experimentally:

-An important conclusion reached by the authors is that Ubp2 antagonizes Mdm30 by directly removing the K48-linked ubiquitins added by Mdm30. To show this, they use an *in vitro* deubiquitination assay with recombinant Ubp2 and 5x ubiquitin as a substrate. They observe that Ubp2 can remove one ubiquitin and thus conclude that, contrary to previous knowledge, Ubp2 can hydrolyze K48 bonds provided that the chain is at least 5 Ubiquitin long. The specificity of the reaction is not very well controlled: basically the authors rely on the redox sensitivity of Ubp2 to conclude that the cleavage is indeed performed by Ubp2, and not by a contaminating protease. To confirm the "minimum 5 Ub" models, the authors should repeat the experiments of figure 2C with longer UB chains (several groups have used K48-linked hexa-ubiquitin, it should thus be commercially available). This should lead to the appearance of Ub5 and Ub4 species. This is important to show that Ub5 is indeed the limit and that the removal of one ubiquitin from the Ub5 observed here is not simply a spurious phenomenon.

-The *in vitro* fusion assays are very tricky. Mitochondria can stick to each other without fusing. Light microscopy will not make any difference, and lipid composition is likely to affect the stickiness of mitochondria. Mitochondrial fusion is however completely blocked by either GMPPNP or CCCP. It is thus worth adding this control to increase the confidence that what is measured here is indeed fusion and not mere stickiness.

-It would be good to have higher exposure panels of all Fzo1 blots, in order to see the ubiquitinated species.

Here are some points that need to be softened or rewritten:

-Because Mdm30 is needed to observe the polyubiquitylated forms of Fzo1 (that are substrates for Ubp2), the author concludes that Mdm30 is directly responsible for generating them. An alternative model is that Mdm30-mediated K48-linked oligoubiquitylation of Fzo1 serves as a signal for the recruitment of a yet-unknown E3-ligase that is going to K63-polyubiquitylate Fzo1, thus

generating ubiquitin forms that are substrates of Ubp2. This model is slightly more complicated than that of the authors, but cannot be excluded here. Thus softer wording and acknowledgement of alternative models is required here.

-The coIP experiment is not well explained at all. F-box is not defined in the main text. Is interaction only observable with an f-box mutant? What happens with the WT protein? Why are the same panels used in the supplement and the main figure?

-"Since mitochondria are separated from each other in vitro, this spatio-temporal regulation may not be recapitulated, unless Fzo1 levels get increased further". This is a bizarre argument that I cannot understand. Fzo1 levels are naturally increased in oleate-treated yeast. Why should Fzo1 levels be increased further? Mitochondria are not "separated" since they are concentrated by centrifugation.

-"This specific increase in mitofusin levels upon treatment with UFAs was accompanied by decreased detection of the doublet characteristic of Fzo1 ubiquitylation by Mdm30 (Fig. 5d; Fzo1-Ub), which is in agreement with the down-regulation of Mdm30-mediated turnover of Fzo1 by stabilized Ubp2 (Fig. 5b)." This argument appears to contradict everything that has been said and shown before. Ubp2 stabilization should have no effect whatsoever on the short-Ub chain doublet since Ubp2 is said not to act only on long Ub chains.

-The model is that Fzo1 levels are regulated to match lipid insaturation levels. In that case deleting Mga2 (thus decreasing insaturation) should lead to decreased Fzo1 levels. Is it the case? (it may look like it in figure 7G, but that should require repetition and quantification). What is the interpretation if it is not the case? It would be good to see higher exposures of these blots so as to be able to see the ubiquitylated products of Fzo1.

-In general, many sentences could be simplified. e.g.

"Absence of the Mdm30 ubiquitin ligase would therefore not restrict to promoting accumulation of mitofusins but may also affect the myriad of functions regulated by the Rsp5/Ubp2 complex."

"These results confirm the specific effect of Mdm30 on Ubp2 over dysfunctional mitochondria and suggest Ubp2 modification by the Mdm30 ubiquitin-ligase."

Reviewer #3 (Remarks to the Author)

This study addresses the functional interplay of the deubiquitinase (DUB) Ubp2 with ubiquitin ligases, Mdm30 and Rsp5. In the first part of the manuscript the authors report that Ubp2 acts as an antagonist for Mdm30-mediated turnover of Fzo1 and that Mdm30 targets Ubp2 for degradation. Only the latter represents a truly novel finding. The Mdm30-dependent reduction of Ubp2 levels promotes Rsp5-dependent induction of Ole1-dependent desaturation of fatty acids. Lack of Mdm30 impairs Ole1-dependent synthesis of desaturated fatty acids. Interestingly, desaturated fatty acids impair degradation of Ubp2 and thereby stabilize Fzo1 and promote mitochondrial fusion. This second part of the manuscript thus reports on the novel link between mitochondrial fusion and fatty acid desaturation which, in my eyes, is a main and highly relevant finding. Overall, the manuscript is well written and the conclusions are largely justified. Still, some experiments are merely confirmatory or have critical problems as they lack important controls or are not clearly explained. Alternative explanations need to be discussed as well. These points need to be resolved before the manuscript is recommended for publication.

Major points:

1. The fact that Ubp2 has an influence on the level/turnover of Fzo1 is already known (Anton et al 2013). The novelty is more about the antagonism to Mdm30. The confirmatory nature and the novel aspect must be made clearer.
2. The authors show that overexpression of Rsp5 rescues defects caused by lack of Mdm30 and conclude that "accumulation of Ubp2 induces downregulation of Rsp5-mediated functions...". This

a possibility but an alternative explanation is that Rsp5 is directly affected by Mdm30. Figure 4a suggests that Rsp5 levels are slightly reduced by loss of Mdm30. Thus, Rsp5-mediated functions are not necessarily affected via Ubp2. The authors should determine whether Mdm30 affects the Rsp5/Ubp2 pathway directly by affecting Rsp5 as well. The authors could test the turnover (CHX chase) of Rsp5 in delta Mdm30.

3. Page 17/18: The rationale for the last part of the results section is not fully clear to me. Why is decreased desaturation and increased turnover of Fzo1 hypothesized to promote fusion? Moreover, the authors state that introduction of an extra copy of Fzo1 induced rescue of glycerol growth and tubular mitochondrial morphology in delta ubp2 and double mutants (delta ubp2 delta mga2). Mitochondrial morphology is, however, not rescued at all (Fig. 7de). This needs to be tested statistically to make such a claim. In lines 22/23 the authors mention "increased levels of mitofusins". This is not observed making this conclusion/sentence unclear to me.

In general, the authors suggest that the turnover of Fzo1 rather than the level is critical for mitochondrial fusion (and balance of UFAs). The author neglect the possibility that decreased amounts of desaturated fatty acids are the main problem and that Fzo1 may simply need desaturated fatty acids for fusion (which implies that UFAs are upstream of Fzo1 turnover).

4. The authors must improve the clarity and the proper labelling of several figures. Several experiments also lack loading controls and/or statistical testing. For specific points see below.
Minor points:

1. Anton et al 2013 reports on Ubp2-dependent species of ubiquitylated Fzo1 that are low in molecular weight (using expression of an inactive variant of Ubp2). These species are not resolved here (e.g. Fig 2b). It would be interesting to see whether these species also do depend on Mdm30 (in addition to the "smear" at high MW).

2. The accumulation of high MW ubiquitylated Fzo1 species in delta Ubp2 are not fully convincing. For Fig. 1c a quantification ($n \geq 3$ experiments) and a loading control is needed.

3. Fig 1d: Are the differences in Fzo1 turnover really significant? Please include statistical tested significance values in Fig 1d.

4. Correct "Page xx on 47" to "Page xx of 47"

5. Page 4 /line 18: "unexplained" is not correct as models exist. Better: "not fully understood".

6. Page 6 / line 18: Explain better why the expression of Spt23 p90 is needed. The strain must not be labeled as delta strain in the figures as the delta is inviable.

7. Page 10 / line 19: "mitochondrial deficiency" is not correct! Respiratory deficiency?

8. Page 11 / line 22: It is not clear why this should be the "sole candidate". I suggest to rephrase this.

9. Page 13 / line 15: "endogenous" is not correct as it is Ubp2-HA.

10. Figure 2b and 2c: Loading controls are missing.

11. In Figure 2d WT and ubp2 Δ Fzo1-Ub banding pattern seems to be different to earlier results (doublet is not clearly seen and high MW species are not enhanced when loading control is taken into account). The immunoblotting is not really convincing. This should be improved. What is the role of K464 residue? Is there an effect in the ubp2 Δ strain with K464R, or K398RK464R?

12. Figure 3: Figure 3a and 3b: Anton et al., 2013 reported previously the growth defect of ubp2 Δ . For the double mutation of Ubp2 and Mdm30 a confirmation including a rescue of the growth defect would be needed. Also a rescue of the growth defect with a plasmid encoding Mdm30 is missing. The labeling of Figure 3c and 3e are unclear/incomplete also the appropriate size of the bands are missing. The authors should indicate that they show that the DUB bind to Mdm30 f-box mutant NOT to Mdm30 WT (as implied here).

13. Figure 4c: Was the quantification of number of zygotes only performed once? Error bars are missing and/or the number of cells analyzed.

14. Western blot of 4e is not convincing and does not fit to the quantification shown. Please improve and indicate size markers.

15. Was the experiment in figure 5a only performed once? Error bars? 5b a quantification of the increase of the protein levels would be helpful. It would be interesting to know whether the Mdm30 expression levels increased or decreased also after UFA treatment in WT and ubp2 Δ cells. 5b and 5d the protein sizes are missing. Significance of 5d (3 vs.4) should be tested.

16. The labelling for Figure 6a is absent (size?, channel?). Figure 6b one time experiment? Error

bars? 6c, statistical analysis should be added for significance.

17. Figure 7, a rescue of the growth defect with a plasmid expressing the protein to verify that the growth defect is specific should be added. Immunoblots with protein size markers are missing (see also remarks above for Figure 7df).

18. Figure 7b: Why is OLE1 expression reduced in delta Ubp2? Please comment.

19. Suppl Fig. 6. Please add experiment without OA.

20. Suppl Fig. 7. Explain what is meant by X. Which dilutions of which strains.

Responses to Reviewers' Comments

To accompany the point-by-point response to Reviewers' comments, below please find an overview of the main changes that have been incorporated into the study.

Overview of main changes

New data:

- Figure 1d: CHX chase of endogenous Fzo1 in *rsp5Δ* cells in response to R3.8
- Figure 2d: *In vitro* deubiquitylation assay with Ub(3-7)K48 chains in response to R2.3
- Figure 4c: *In vivo* fusion assay with *rsp5Δ* cells in response to R1.8 and R3.18
- Figure 5a: *In vivo* fusion assay with WT cells treated with Oleic Acid in response to R3.20
- Figure 5c: Effect of Oleic Acid treatment on Fzo1 ubiquitylation in response to R2.9
- Figure 8a: CHX chase of endogenous Fzo1 in *mga2Δ* cells in response to R2.10
- Supplementary Figure 2d: *In vitro* deubiquitylation assay with Ub(3-7)K63 chains in response to R2.3
- Supplementary Figure 3a: Spot assay with *ubp2Δ* cells expressing WT *UBP2* in response to R3.17 and R3.22
- Supplementary Figure 4b: CHX chase of endogenous Rsp5 in response to R3.3
- Supplementary Figure 5a: Effect of Oleic Acid treatment on Mdm30 levels in response to R3.20
- Supplementary Figure 6b: *In vitro* fusion assay upon treatment with CCCP or GMP-PNP in response to R2.4
- Supplementary Figure 8a: Ole1 protein levels in *ubp2Δ* cells in response to R3.23
- Supplementary Figure 8b: Spot assay with *ubp2Δ mga2Δ* clone 1 expressing WT *FZO1* in response to R3.25
- Figure A for reviewers: Fzo1 ubiquitylation in *mga2Δ* cells in response to R2.10, R3.15
- Figure C for reviewers: Spot assay with *mdm30Δ* expressing WT *FZO1* without Oleic Acid in response to R3.24
- Figure D for reviewers: Spot assay with *mdm30Δ* expressing *fzo1 K464R* in response to R3.16 and R3.24

Modification of existing Figures:

- Figure 1c, Figure 2e: contrast of long expo blots were enhanced in response to R3.7 and R3.16
- Figures 2a, 3f and supplementary 1b, 1c: error bars were changed from s.d. to s.e.m. in response to R3.8
- Figure 3c and 3e: reorganized in response to R3.17
- Figures 1c, 2b and supplementary 1a: "Loading normalized to Fzo1-13Myc levels" added in response to R3.7 and R3.15
- Figure 3d: Whole Figure moved from supplementary data to the main Figures in response to R2.7 and R3.17
- Figures 3a, 3b, 3g, 4b and supplementary 3b, 3d, 4a: "MDM30 shuffling strains" added in response to R3.17
- Figure 4e: reorganization and statistical testing were performed in response to R3.19
- Figure 5b: Quantification was performed in response to R3.20
- Figures 5d, 6c, 7d and 8c: Statistical testing were performed in response to R3.4, R3.20 and R3.21
- Figure 6a: The labelling was completed in response to R3.21
- Figure 7c, supplementary 8b: X was replaced by the name of the strains in response to R3.25
- Figure 8e: Figure moved from supplementary data to the main Figures in response to R3.4

Text editing: The main text was edited to comply with reviewers suggestions and to comment on new data added in the manuscript. Most changes are notified in the responses below and original word files with track changes have been submitted together with the revised study.

We thank Reviewers for their assessment of our manuscript. We hope that with the new experiments and added clarifications in the revised study, our work will be viewed as presenting original and provocative findings that represent a major advance in our understanding of mitochondrial fusion regulation, suitable for publication in Nature Communications.

Point-by-point responses to reviewers' comments

Reviewer #1 (Remarks to the Author):

The manuscript by Cavellini, et al., describes relationships between Ubp2 (a deubiquitinating enzyme), Fzo1 (a mitofusin), and two ubiquitin ligases (Mdm30 and Rsp5). Many of the central conclusions are poorly substantiated and there is a lack of molecular understanding of several of the key observations. Given this, the contribution of the work to the field appears to be minimal. Also, the manuscript is in need of extensive editing. The meaning of several key statements is nearly completely lost due to poor sentence structure and word choice.

Response R1.1: We thank the reviewer for evaluating our manuscript but regret that this summary makes abstraction of the new insights that the study brings on mitochondrial fusion.

Specific comments:

It is unclear why the authors use the word "canonical" when referring to Ubp2 as a "canonical antagonist" of Rsp5. While Ubp2 is an antagonist of Rsp5, their relationship/interaction/mechanism does not follow any general rule and is therefore not canonical of anything.

Response R1.2: "... a canonical antagonist of Rsp5..." has been replaced by "... an antagonist of Rsp5..." (Page 4, line 21 and page 6, line 15).

Figure 1: After several readings, I still have no understanding of what is meant by Fzo1 "short" and "long".

Response R1.3: As mentioned in the Figure legends, "short" and "long" meant short or long exposures of immunoblots. In the revised manuscript, "short" and "long" have been changed to "short expo" and "long expo".

Page 7, first paragraph: an example of the confusion in the writing. It is first said that the "absence of Mdm30 stabilized the mitofusin whether or not UBP2 was also deleted", then two sentences later it is stated that "these results indicate that the accelerated turnover of Fzo1 that is induced by the absence of Ubp2 is abolished in the absence of Mdm30." The first statement is simple and direct, while the second implies some type of regulatory "order" to the system that simply does not exist in either the experiment or in nature.

Response R1.4: The second sentence has been changed to "Mdm30 is essential for the accelerated turnover of Fzo1 that is induced by the absence of Ubp2" (Page 7, line 18). This statement, or even the former, does not imply any type of regulatory order but simply reflect that while Fzo1 turnover is accelerated in ubp2Δ cells, this is no longer the case when Mdm30 is absent (see Figure 2a).

The section on K63 versus K48 chain type specificity of Ubp2 is confusing. The in vitro results are consistent with Ubp2 having very low activity against K48 chains (and perhaps no activity against K48 chains in cells, but this is difficult to know). However, the results are interpreted as being consistent with the fact that Mdm30 catalyzes K48 chain formation on Fzo1. This makes seem to make no sense. The types of chains formed on Fzo1 by Mdm30 must be addressed by determining the type of polyubiquitin chains that are accumulating on Fzo1 in the ubp2 deletion mutant - either by mass spec of the purified protein or perhaps by analysis in a K63R yeast

strain. Also, while it is shown that all ubiquitylation of Fzo1 is dependent on K398, I don't see how this bears on the question of chain type. The general issue here is that if Mdm30 is directly ubiquitylating Fzo1 and Ubp2 is directly reversing this reaction, then Mdm30 and Ubp2 are expected to be synthesizing and removing, respectively, the same type of chains.

Response R1.5: The section on K63 versus K48 chain specificity of Ubp2 relies on previous findings by Anton et al. 2013. In this study, Ubp2 was shown to specifically interact with high molecular weight species of Fzo1. Moreover, the types of chains on Fzo1 immunoprecipitated from WT and ubp2Δ cells were analyzed by mass spectrometry. The result of this experiment was unequivocal: Fzo1 is essentially modified with K48 linkages in WT cells and this specific kind of linkages significantly increases in the absence of Ubp2. While it was not evaluated further in Anton et al., the most straightforward interpretation from this set of results is that Ubp2 antagonizes K48-ubiquitylation of Fzo1.

Consistent with this, we show that Rsp5, which is a K63 E3, is not involved in the regulation of Fzo1 ubiquitylation and degradation by Ubp2 (Figure 1) but that this function rather involves Mdm30, which is a K48 E3 (Figure 2). We then address the differential capacity of Ubp2 to disassemble K63 and K48-linked chains. Figure 2c and the new Figure 2d indicate that while Ubp2 has the capacity to disassemble K63-linked chains independent of their size, its activity on K48-linked chains is not null but restricted to longer chains composed of 5 Ub moieties or more. This observation together with those from Anton et al. suggest that the high molecular weight species of Fzo1 that accumulate in the absence of Ubp2 may correspond to K48-Ub chains that extend from the K48-Ub doublet which is seen in WT cells (see Figure 2b). However, Fzo1 contains 78 lysine residues in its sequence and one can therefore not exclude that chains get added on lysines distinct from K398, on which the Ub doublet is conjugated. The demonstration that all ubiquitylation of Fzo1 is dependent on K398, does not bear on the question of chain type but relates to the confirmation that, in the absence of Ubp2, Ub chains may extend from the doublet conjugated on K398 of Fzo1.

We hope these precisions will help clarifying the conclusions than can be drawn from the dataset shown in Figure 2.

The section entitled "Ubp2 is a physiological substrate for Mdm30-mediated degradation" is underwhelming and incomplete. The changes in Ubp2 levels appear marginal, as does the co-IP of Mdm30 and Ubp2. Also, the types of chains that are accumulating on Ubp2 should again be examined. Are these really K48 chains, as would be expected if catalyzed by Mdm30, or are they K63 chains (perhaps catalyzed by its association with Rsp5). It is unclear whether attempts were made at in vitro reconstitution of the interactions between Mdm30 and Ubp2 and Fzo1, but this is what is clearly needed to get any molecular insight into what is going on here.

Response R1.6: The changes in Ubp2 levels are qualified as marginal but these changes are totally reproducible (seen in Figures 3c, 3d and 3e) and the slow degradation rate of Ubp2 (Figure 3f) actually explains the only two-fold increase of Ubp2 upon stabilization. We are truly sorry that the co-IP between Mdm30 and Ubp2 (Figure 3d) is also qualified as marginal. This is a subjective comment to which no rational response can be provided. We can only reiterate the fact that Ubp2 does co-immunoprecipitate with the F-box mutant of Mdm30 and that the lack of co-IP with WT mdm30 is consistent with the highly transient interactions that usually take place between substrates and E3s, especially when these interactions result in degradation of the substrate. We do not question that analyzing the types of chains conjugated to Ubp2 could be interesting and informative. However, this goes beyond the scope of this already complex study as it is not essential to draw the conclusion

that Mdm30 regulates the turnover of Ubp2. Far more important in this matter is the observation that high molecular weight species of Ubp2 decrease in the absence of Mdm30 (Figure 3e). Regarding the in vitro reconstitution of the interactions between Mdm30, Ubp2 and Fzo1, this is unfortunately not feasible at this point because purification of recombinant Fzo1 remains a major roadblock in the field.

Page 11, lines 8-12: The conclusions made in the last two sentences of this paragraph are vague and confusing. "Convey" is a poor word choice - should this be replaced by suggest/imply, or demonstrates/indicates? The last sentence of the paragraph is worse, as I am unclear what the evidence is that Mdm30 affects the myriad activities of Rsp5.

Response R1.7: The word "convey" has been replaced by the word "suggest" (Page 12, line15) and the last sentence of the paragraph has been deleted.

Figure 4c (mitochondrial fusion assay): It is entirely unclear why the authors found it surprising that there was total co-localization of fluorophores in the rsp5-delta strain. There are absolutely no controls for this experiment - just one highly engineered experimental strain was examined (rsp5-delta expressing Spt23 "p90").

Response R1.8: Given the pleiotropy of Rsp5 and its established involvement in regulation of mitochondrial homeostasis, we were indeed surprised to find that mitochondrial fusion is not affected in cells lacking Rsp5. The word "surprisingly" has nonetheless been deleted. It is unclear whether qualifying the strain we employed as a highly engineered one must be interpreted as a criticism. The point is that deletion of RSP5 is inviable unless the N-terminal fragment of Spt23 remains expressed in trans. This experiment has been repeated and WT controls have been added (New Figure 4c). In all cases, 100% fusion efficiency was observed indicating that upon maintenance of the OLE1-pathway, mitochondrial fusion remains fully functional in the absence of Rsp5.

Reviewer #2 (Remarks to the Author):

Herein, Cavellini and colleagues describe the antagonistic effects of the deubiquitylase enzyme Ubp2 and of the ubiquitin ligase Mdm30 on the mitofusin Fzo1 in yeast and on mitochondrial function. While Ubp2 deletion destabilizes Fzo1, Mdm30 deletion stabilizes it. They then describe an interesting cross talk between the Ole1 pathway of lipid desaturation (in which Ubp2 plays a role) and mitochondrial fission, indicating that Fzo1 levels must be tightly regulated in the face of fluctuations in lipid saturation.

The study is original, unexpected and interesting.

The data are in general of excellent quality and support the most important conclusions of the study. It is a bit unfortunate that the authors did not go as far as testing that the various treatments indeed affect mitochondrial lipid composition as expected, but this is not crucial at this point.

Response R2.1: We thank the reviewer for this accurate overview of the findings presented in the study.

Some additional work and some rewriting could nevertheless improve the manuscript.

Here are the points that need to be addressed experimentally:

-An important conclusion reached by the authors is that Ubp2 antagonizes Mdm30 by directly removing the K48-linked ubiquitins added by Mdm30. To show this, they use an in vitro deubiquitination assay with recombinant Ubp2 and 5x ubiquitin as a substrate. They observe that Ubp2 can remove one ubiquitin and thus conclude that, contrary to previous knowledge, Ubp2 can hydrolyze K48 bonds provided that the chain is at least 5 Ubiquitin long. The specificity of the reaction is not very well controlled: basically the authors rely on the redox sensitivity of Ubp2 to conclude that the cleavage is indeed performed by Ubp2, and not by a contaminating protease.

Response R2.2: We need here to emphasize that our experimental setup is designed to exclude such possibility that other protease than Ubp2 triggers the in vitro disassembly of Ubiquitin chains. We worked with Ubp2-HA immunoprecipitated from genomically tagged UBP2-HA cells. The negative control here consists in immunoprecipitates from UBP2-HA lysates obtained in the absence of HA antibody (now mentioned in the main text; page 8, line 7) and shows that the basal status of Ubiquitin chains is unaffected. The status of K63 and K48-linked chains only changed when HA antibodies were added and Ubp2-HA was therefore present in the reactions. One could argue that the ubiquitin protease activities seen with Ubp2-HA are due to a distinct ubiquitin-protease that co-precipitates with Ubp2. However this is very unlikely not only because such a DUB interacting physically with Ubp2 has never been documented but also because the strong responses to redox variations we obtained do perfectly match with the ubiquitin-protease activity of Ubp2.

To confirm the "minimum 5 Ub" models, the authors should repeat the experiments of figure 2C with longer UB chains (several groups have used k48-linked hexa-ubiquitin, it should thus be commercially available). This should lead to the appearance of Ub5 and Ub4 species. This is important to show that Ub5 is indeed the limit and that the removal of one ubiquitin from the Ub5 observed here is not simply a spurious phenomenon.

Response R2.3: Overall we do fully agree with this very pertinent comment. To perform this experiment, we searched for commercially available hexa-ubiquitin chains. However, we were unable to find any vendor selling K48-linked hexa chains. The only product we found that could be compatible with the experiment consists in

mixes of chains composed of 3 to 7 Ub moieties assembled through either K63 or K48 linkages. Repeating the experiment with these substrates demonstrated that Ubp2 could cleave all K63-chains independent of their length to yield di-Ub products (New Supplementary Figure 2d). In contrast, Ubp2 could only disassemble Ub7 and Ub6K48 chains to generate Ub3K48 products (New Figure 2d). A possible effect on Ub5 or Ub4K48 chains was more difficult to detect probably because of Ub5 and Ub4 byproducts resulting from Ub7 and Ub6 disassembly. The main text has been edited to comment on this new set of data (Page 8, line 18 ... Page 9, line 4). These results together with those obtained with Ub5K48 chains (Figure 2c) strongly suggest that Ubp2 could remove as much as 3 ubiquitin moieties at a time from long K48-linked chains (Ub7 and Ub6) and 1 moiety from shorter Ub5K48 chains. Most importantly, we hope the reviewer will agree that we now confirm that while Ubp2 can disassemble K63-linked chains independent of their size, its K48-trimming activity is restricted to longer ubiquitin chains.

-The in vitro fusion assays are very tricky. Mitochondria can stick to each other without fusing. Light microscopy will not make any difference, and lipid composition is likely to affect the stickiness of mitochondria. Mitochondrial fusion is however completely blocked by either GMPPNP or CCCP. It is thus worth adding this control to increase the confidence that what is measured here is indeed fusion and not mere stickiness.

Response R2.4: In in vitro mitochondrial fusion reactions, light microscopy cannot differentiate intermediates that fused their outer membrane from mitochondria that are attached together. This is only true if both the GFP and the mCherry signals emanate from mitochondrial matrices. In our setup, this not the case as GFP is expressed on the outer membrane (OM45-GFP) whereas mCherry lies in the matrix (mito-mCherry). In this context light microscopy clearly differentiates attached mitochondria that appear as proximal GFP-mCherry signals from outer membrane fused intermediates that appear as colocalized GFP-mCherry signals. While this is actually shown in one of our recent publications (Brandt et al. 2016), we agree with the reviewer that adding controls in which colocalization decreases when outer membrane fusion is inhibited would clearly bring more confidence and reinforce our observations. As proposed by the reviewer, the new Supplementary Figure 6b demonstrates that colocalization (and therefore outer membrane fusion) decreases upon treatment with CCCP or GMP-PNP. Notably, these effects totally corroborate those published previously in Meeusen et al. 2004 and consequently provide strong validation of our initial data. The main text has been edited to comment on this new data (Page 16, lines 1-4).

-It would be good to have high exposure panels of all Fzo1 blots, in order to see the ubiquitinated species.

Response R2.5: Fzo1 tagging is essential to detect ubiquitylation of the mitofusin because our anti-Fzo1 is unfortunately not sensitive enough to detect high MW species of the mitofusin. Notably, however, the ubiquitylation status of tagged Fzo1 has been assessed in all mutants and in all conditions used in the study (see in particular new Figure 5c in the presence or in the absence of Oleic Acid treatment) with the exception of mga2Δ cells (see response R2.10).

Here are some points that need to be softened or rewritten:

-Because Mdm30 is needed to observe the polyubiquitylated forms of Fzo1 (that are substrates for Ubp2), the author concludes that Mdm30 is directly responsible for generating them. An alternative model is that Mdm30-mediated K48-linked oligoubiquitylation of Fzo1 serves as a signal for the recruitment of a yet-unknown E3-ligase that is going to K63-polyubiquitylate Fzo1, thus generating ubiquitin forms that are substrates of Ubp2.

This model is slightly more complicated than that of the authors, but cannot be excluded here. Thus softer wording and acknowledgement of alternative models is required here.

Response R2.6: This is a good point with which we agree. Anton et al. have analyzed Fzo1 ubiquitylated species in WT and ubp2Δ cells by mass spectrometry and have shown that Fzo1 undergoes strong increase in K48 but not K63 ubiquitylation upon deletion of UBP2. Consequently, extension of the Mdm30-dependent doublet that we detect in ubp2Δ cells likely corresponds to K48 rather than K63 ubiquitylation. These considerations are now stated in the main text (Page 7, lines 3-5). It is therefore correct to state that Mdm30 primes K48-ubiquitylation on Fzo1 but that while the extension of the doublet with K48-linked chains may be mediated by Mdm30, the involvement of a distinct E3 cannot be excluded at this stage. To reflect this, a new paragraph has been added at the end of Page 9 (starting line 15) and the main text has been edited by replacing "...Mdm30 is the E3..." with "...Mdm30 is essential for..." inducing the faster turnover of Fzo1 in the absence of Ubp2 (Page 8, line 3 and Page 10, line 6).

-The coIP experiment is not well explained at all. F-box is not defined in the main text. Is interaction only observable with an f-box mutant? What happens with the WT protein? Why are the same panels used in the supplement and the main figure?

Response R2.7: This comment is well taken. We now emphasize that "a non-functional mutant version in the F-box motif of Mdm30-Myc is unable to bind other SCF components" (Page 10, line 16). We initially placed the whole co-IP figure and its full explanation in supplement to limit the length of the main text. We apologize if this led to loss of information. The partial figure has now been removed and replaced by the whole figure (Figure 3d) and the full explanation of the figure has been included in the main text (Page 10, line 21 ... Page 11, line 6).

-"Since mitochondria are separated from each other in vitro, this spatio-temporal regulation may not be recapitulated, unless Fzo1 levels get increased further". This is a bizarre argument that I cannot understand. Fzo1 levels are naturally increased in oleate-treated yeast. Why should Fzo1 levels be increased further? Mitochondria are not "separated" since they are concentrated by centrifugation.

Response R2.8: We agree that this sentence is difficult to understand and does not really reflect what we initially wanted to express. The main text has now been extensively edited to better relate our line of thoughts (see Page 16, lines 7-14).

-"This specific increase in mitofusin levels upon treatment with UFAs was accompanied by decreased detection of the doublet characteristic of Fzo1 ubiquitylation by Mdm30 (Fig. 5d; Fzo1-Ub), which is in agreement with the down-regulation of Mdm30-mediated turnover of Fzo1 by stabilized Ubp2 (Fig. 5b)." This argument appears to contradict everything that has been said and shown before. Ubp2 stabilization should have no effect whatsoever on the short-Ub chain doublet since Ubp2 is said not to act only on long Ub chains.

Response R2.9: We do thank the reviewer for bringing this up. To address this issue, we have now analyzed the in vivo ubiquitylation status of tagged Fzo1 in the absence or in the presence of Oleic Acid treatment. This new data shows that the increase in Fzo1 levels upon addition of Oleic Acid is accompanied by a 50% increase of the ubiquitylation doublet (New Figure 5c). As expected, these effects are abolished upon deletion of UBP2 and MDM30. These results now fully agree with those of Figure 2 as stabilized Ubp2 likely blocks extension of the Fzo1 ubiquitylation doublet resulting in stabilization of the mitofusin. The main text has been edited to comment on this new data (Page 15, lines 8-14). The high exposure of the anti-Fzo1 blot from former Figure 5d

has otherwise been removed as the new Figure 5c strongly suggests that loss of the Fzo1 ubiquitylation doublet on mitochondria purified from Oleic Acid treated cells likely occurred during mitochondrial isolation.

-The model is that Fzo1 levels are regulated to match lipid insaturation levels. In that case deleting Mga2 (thus decreasing insaturation) should lead to decreased Fzo1 levels. Is it the case ? (it may look like it in figure 7G, but that should require repetition and quantification). What is the interpretation if it is not the case? It would be good to see higher exposures of these blots so as to be able to see the ubiquitylated products of Fzo1.

Response R2.10: This is a very good point! Mitochondrial fusion (as assessed by glycerol growth and mitochondrial morphology) is not affected in mga2Δ cells that are characterized by notably low desaturation of fatty acids. According to our findings, and as noticed by the reviewer, Fzo1 levels should somehow be impacted in these cells. However, the steady state levels of endogenous Fzo1 and the ubiquitylation status of tagged Fzo1 are not affected upon deletion of MGA2 whatsoever (see accompanying Figure A for reviewers). This led to monitor the turnover of endogenous Fzo1 in WT and mga2Δ cells. As expected from the Fzo1/Ole1 balance, we found that degradation of Fzo1 is slightly more rapid in the absence than in the presence of Mga2 (New Figure 8a). The only rational explanation resulting from these observations is that diminished fatty acid desaturation induces an upregulation of Fzo1 expression which compensates faster turnover of the mitofusin. Importantly, this also provides straightforward consistency with the full rescue in glycerol growth and mitochondrial morphology of ubp2Δ mga2Δ cells upon addition of the FZO1 extra-copy. The main text has been edited to reflect these considerations (Page 19, lines 9-20).

-In general, many sentences could be simplified. e.g.

"Absence of the Mdm30 ubiquitin ligase would therefore not restrict to promoting accumulation of mitofusins but may also affect the myriad of functions regulated by the Rsp5/Ubp2 complex."

Response R2.11: Agreed. This sentence has been removed.

"These results confirm the specific effect of Mdm30 on Ubp2 over dysfunctional mitochondria and suggest Ubp2 modification by the Mdm30 ubiquitin-ligase."

Response R2.12: Agreed. This sentence has been edited (Page 11, lines 11-14).

Reviewer #3 (Remarks to the Author):

This study addresses the functional interplay of the deubiquitinase (DUB) Ubp2 with ubiquitin ligases, Mdm30 and Rsp5. In the first part of the manuscript the authors report that Ubp2 acts as an antagonist for Mdm30-mediated turnover of Fzo1 and that Mdm30 targets Ubp2 for degradation. Only the latter represents a truly novel finding. The Mdm30-dependent reduction of Ubp2 levels promotes Rsp5-dependent induction of Ole1-dependent desaturation of fatty acids. Lack of Mdm30 impairs Ole1-dependent synthesis of desaturated fatty acids. Interestingly, desaturated fatty acids impair degradation of Ubp2 and thereby stabilize Fzo1 and promote mitochondrial fusion. This second part of the manuscript thus reports on the novel link between mitochondrial fusion and fatty acid desaturation which, in my eyes, is a main and highly relevant finding. Overall, the manuscript is well written and the conclusions are largely justified.

Response R3.1: Many thanks for this accurate summary.

Still, some experiments are merely confirmatory or have critical problems as they lack important controls or are not clearly explained. Alternative explanations need to be discussed as well. These points need to be resolved before the manuscript is recommended for publication.

Major points:

1. The fact that Ubp2 has an influence on the level/turnover of Fzo1 is already known (Anton et al 2013). The novelty is more about the antagonism to Mdm30. The confirmatory nature and the novel aspect must be made clearer.

Response R3.2: The reviewer is absolutely correct. We do apologize if this distinction between what was initially found in Anton et al. and our original contributions in Figures 1 and 2 did not turn as explicit as expected. We thus added a new paragraph in the results section that clearly delineates the respective contributions of both studies (Page 9; lines 15-21). We would also like emphasizing that all original discoveries from Anton et al. are reminded throughout the study: in the introduction (Page 4, lines 12-16 and 19-20), in the results section (Page 6, lines 3-4; Page 7, lines 4-5, 8-9 and 13-14; Page 8, lines 1-2; Page 9, lines 12-13; Page 10, lines 1-2; Page 18, line 15) and in the discussion (Page 21, lines 2-14).

2. The authors show that overexpression of Rsp5 rescues defects caused by lack of Mdm30 and conclude that "accumulation of Ubp2 induces downregulation of Rsp5-mediated functions...". This a possibility but an alternative explanation is that Rsp5 is directly affected by Mdm30. Figure 4a suggests that Rsp5 levels are slightly reduced by loss of Mdm30. Thus, Rsp5-mediated functions are not necessarily affected via Ubp2. The authors should determine whether Mdm30 affects the Rsp5/Ubp2 pathway directly by affecting Rsp5 as well. The authors could test the turnover (CHX chase) of Rsp5 in delta Mdm30.

Response R3.3: In the initial submission we could already exclude this possibility that Mdm30 acts on Rsp5 degradation as the CHX chase that is suggested to be performed was actually shown in supplementary data (see supplementary Figure 2b in the revised manuscript). We agree however that this control experiment is absolutely essential to support our conclusions and therefore deserves more emphasis than initially attributed. Consequently this CHX chase has been repeated and quantified (New supplementary Figure 4b) and demonstrates that Rsp5 turnover is not affected in the absence of Mdm30. The main text has been edited to bring more emphasis on this important control (Page12, line 14).

3. Page 17/18: The rationale for the last part of the results section is not fully clear to me. Why is decreased desaturation and increased turnover of Fzo1 hypothesized to promote fusion? Moreover, the authors state that introduction of an extra copy of Fzo1 induced rescue of glycerol growth and tubular mitochondrial morphology in $\Delta ubp2$ and double mutants ($\Delta ubp2 \Delta mga2$). Mitochondrial morphology is, however, not rescued at all (Fig. 7de). This needs to be tested statistically to make such a claim. In lines 22/23 the authors mention "increased levels of mitofusins". This is not observed making this conclusion/sentence unclear to me.

Response R3.4: We apologize if this last part of the study appeared unclear to the reviewer. A key aspect of this section is the observation that the FZO1 extra copy induces total rescue of glycerol growth in the $ubp2\Delta mga2\Delta$ double mutant but has no effect on the $ubp2\Delta$ single mutant (Figure 8b). This very strong differential effect cannot be attributed to changes in Fzo1 levels or turnover between the two cell types because the absence of Ubp2 induces similarly low levels of Fzo1 whether the FZO1 extra-copy is introduced or not (Figure 8d). In other words, Fzo1 is degraded faster in the absence of Ubp2 and even more Fzo1 molecules are subject to this degradation upon introduction of the FZO1 extra-copy in both single and double mutants. In this context, it is the absence of Mga2 combined with this increased turnover of Fzo1 that allows restoration of respiratory growth in cells that lack Ubp2. Consistent with this, the K398 residue of Fzo1, which is essential for formation of the Mdm30-dependent ubiquitylation doublet (Figure 2e), is also essential for promoting the total rescue of glycerol growth in $ubp2\Delta mga2\Delta$ (data moved from supplementary materials to the main Figure 8e).

We have performed statistical testing on mitochondrial morphology assays. We absolutely agree with the reviewer that mitochondrial morphology in cells that lack Ubp2 is not rescued at all upon deletion of MGA2 (Figure 7d). However, while the introduction of the FZO1-extra copy does not rescue the 15% decrease of tubular mitochondria in the $ubp2\Delta$ single mutant, it does promote total restoration of normal mitochondrial morphology in the $ubp2\Delta mga2\Delta$ double mutant (Figure 8c). This result is thus consistent with those obtained in respiratory growth.

To clarify this section, the former Figure 7 has been split into two distinct Figures according to the absence (New Figure 7) or to the presence (New Figure 8) of the FZO1 extra-copy and the main text has been edited to comment on new experiments added in the revised manuscript (see responses R2.10 and R3.23). The reviewer is otherwise correct about former lines 22/23: "increased levels of mitofusins" has now been replaced by "low levels of mitofusins" (Page 20, line 5).

In general, the authors suggest that the turnover of Fzo1 rather than the level is critical for mitochondrial fusion (and balance of UFAs). The author neglect the possibility that decreased amounts of desaturated fatty acids are the main problem and that Fzo1 may simply need desaturated fatty acids for fusion (which implies that UFAs are upstream of Fzo1 turnover).

Response R3.5: We thank the reviewer for raising this possibility that Fzo1 may simply require desaturated fatty acids to be active. In this context, mitochondrial fusion should be strongly impaired upon decreased fatty acids desaturation. However, both respiratory growth and mitochondrial morphology remain unaffected upon deletion of MGA2. Moreover, we now show that Fzo1 undergoes faster degradation in $mag2\Delta$ cells (New Figure 8a), which agrees with the notion that Fzo1 turnover facilitates mitochondrial fusion upon low desaturation of fatty acids (see also response R2.10).

4. The authors must improve the clarity and the proper labelling of several figures. Several experiments also lack loading controls and/or statistical testing. For specific points see below.

Minor points:

1. Anton et al 2013 reports on Ubp2-dependent species of ubiquitylated Fzo1 that are low in molecular weight (using expression of an inactive variant of Ubp2). These species are not resolved here (e.g. Fig 2b). It would be interesting to see whether these species also do depend on Mdm30 (in addition to the "smear" at high MW).

Response R3.6: The reviewer is absolutely correct. Anton et al. detected the accumulation of Fzo1 high MW species in ubp2Δ cells that migrate below the Mdm30-dependent doublet. These species do not depend on Mdm30 as they persisted in ubp2Δ mdm30Δ cells, which led Anton et al. to propose that Ubp2 does not antagonize Mdm30-mediated ubiquitylation of Fzo1.

In retrospect, it is therefore likely that two distinct modifications of Fzo1 take place: the Mdm30-dependent and the Mdm30-independent high MW species migrating respectively above and below the characteristic ubiquitylation doublet. Our manuscript rather focuses on the first set, which depends on Mdm30 and that appears corresponding to chains that elongate from the doublet initially conjugated to Lysine 398 of the mitofusin. The second set that does not depend on Mdm30, was described in Anton et al. Investigating its nature and function would require a whole set of further investigations that clearly go above the scope of the present study.

2. The accumulation of high MW ubiquitylated Fzo1 species in delta Ubp2 are not fully convincing. For Fig. 1c a quantification (n>=3 experiments) and a loading control is needed.

Response R3.7: While faint, the high MW Fzo1 species that accumulate in the absence of Ubp2, are clearly detectable but, most notably, systematically and totally reproducibly observed regardless of the conditions used (Figure 1b, 1c, 2b, 2e, 5c, supplementary 2c). This brings, in our view, strong consistency and unequivocal confidence in the existence of these species. In Figure 1c, it is the tenuous decrease of Fzo1 high MW species in RSP5 negative (Figure 1c, lanes 2 and 4) as compared to RSP5 positive cells (lanes 1 and 3) that may be misleading. This slight decrease is actually due to the lower levels of Fzo1 in lanes 2 and 4 as compared to lanes 1 and 3. This emphasizes the requirement for normalizing the loading of Fzo1 in order to analyze high MW species of the mitofusin (see response R3.15). In other words, when it comes to analyze the status of Fzo1 ubiquitylation between distinct strains, unmodified Fzo1 is the loading control.

Importantly, the decreased levels of Fzo1 in RSP5 null cells are due to decreased amounts of total proteins in extracts prepared from rsp5Δ cells. This is illustrated by the ponceau staining that is provided for Reviewer 3 (see accompanying Figure B for reviewers). These decreased amounts of total proteins are likely caused by increased cell death during growth of rsp5Δ cells in liquid media.

In Figure 1c, it is thus important to compare lane 1 to lane 3 to see the effect of UBP2 deletion on Fzo1 high MW species and then lane 2 to lane 4 to see that this effect of UBP2 deletion clearly pertains in the absence of Rsp5. This demonstrates that Ubp2 affects Fzo1 ubiquitylation independently of Rsp5. To better emphasize this, the contrast of the long exposure panel in Figure 1c has been enhanced.

3. Fig 1d: Are the differences in Fzo1 turnover really significant? Please include statistical tested significance values in Fig 1d.

Response R3.8: In the original manuscript all error bars in CHX chases corresponded to standard deviations from three independent experiments. In the revised manuscript, these error bars now reflect the standard error of the mean (s.e.m.) for each point, which is actually better adapted for experimental results with a control such as CHX chases. In the particular case of former Figure 1d (Supplementary Figure 1c in the revised manuscript), the point was to demonstrate that the effect of UBP2 deletion on Fzo1 degradation persists in the absence of Rsp5. We have therefore repeated this CHX chase by following endogenous Fzo1 in all strains. We hope the reviewer will agree that this result from new Figure 1d really speaks for itself in that the effect of UBP2 deletion on Fzo1 degradation clearly persists in the absence of Rsp5.

4. Correct "Page xx on 47" to "Page xx of 47"

Response R3.9: Done

5. Page 4 /line 18: "unexplained" is not correct as models exist. Better: "not fully understood".

Response R3.10: Agreed (see page 4, line 18).

6. Page 6 / line 18: Explain better why the expression of Spt23 p90 is needed. The strain must not be labeled as delta strain in the figures as the delta is inviable.

Response R3.11: Done in the main text (page 6, line 18/19) and in Figures (rsp5Δ has been replaced by rsp5Δ + Spt23).*

7. Page 10 / line 19: "mitochondrial deficiency" is not correct! Respiratory deficiency?

Response R3.12: Agreed (see page 12, line 2).

8. Page 11 / line 22: It is not clear why this should be the "sole candidate". I suggest to rephrase this.

Response R3.13: Agreed. The original sentence has been replaced by: "...the OLE1-pathway stood as a remaining candidate possibly contributing to mitochondrial fusion deficiency in mdm30Δ cells." (see page 13, line 12/13).

9. Page 13 / line 15: "endogenous" is not correct as it is Ubp2-HA.

Response R3.14: Agreed. "endogenous" has been replaced by "genomically tagged" (Page 14, line 21).

10. Figure 2b and 2c: Loading controls are missing.

Response R3.15: Figure 1c (see response R3.7), Figure A for reviewers and Figure 2e (see response R3.16) are perfect examples for illustrating the requirement in comparing the pattern of Fzo1 high MW species at constant levels of Fzo1. The mentioning of "Loading normalized to Fzo1-13Myc levels" is specifically used as it indicates that loading of the samples was adjusted to obtain equal levels of unmodified Fzo1 in all lanes to ensure a legitimate comparison of the high MW species pattern in all conditions. As in Figure 1c, unmodified Fzo1 is used as the loading control so that the status of Fzo1 ubiquitylation can be carefully analyzed between distinct

strains. Despite such normalization in Figure 2b, Fzo1-13Myc levels were still lower in *ubp2Δ* cells. High MW species of the mitofusin remained nonetheless significantly stronger in this lane than in the others. In the revised manuscript, “Loading normalized to Fzo1-13Myc levels” has thus been added on Figure 2b. Please note that Figure 2c is a distinct experiment that consists in an *in vitro* deubiquitylation assay. In this assay, equal amounts of Ubp2-HA or Ubiquitin chains have been added in each reaction.

11. In Figure 2d WT and *ubp2Δ* Fzo1-Ub banding pattern seems to be different to earlier results (doublet is not clearly seen and high MW species are not enhanced when loading control is taken into account). The immunoblotting is not really convincing. This should be improved. What is the role of K464 residue? Is there an effect in the *ubp2Δ* strain with K464R, or K398RK464R?

Response R3.16: In former Figure 2d (Figure 2e in the revised manuscript), the loading has not been normalized to Fzo1-13Myc levels. This explains why the relative Fzo1-Ub banding pattern in WT and ubp2Δ is different from results in 1b, 1c and 2b. In particular, the Fzo1-Ub doublet diminishes at the profit of high MW species upon deletion of UBP2, which is consistent with the extension of ubiquitin chains from the doublet. To achieve better visualization of Fzo1-Ub species in the distinct strains, the contrast of the long exposure panel in Figure 2e has been enhanced.

Notably, the main purpose of this Figure lies on the observation that ubiquitylation is no longer detected upon mutation of Lysine 398. This indicates that out of the 78 lysine residues of Fzo1, K398 is the main target of Mdm30 for modification of Fzo1 by the Ub doublet. In this context, ubiquitylation of K464 may serve other functions (see response R3.24).

12. Figure 3: Figure 3a and 3b: Anton et al., 2013 reported previously the growth defect of *ubp2Δ*. For the double mutation of Ubp2 and Mdm30 a confirmation including a rescue of the growth defect would be needed. Also a rescue of the growth defect with a plasmid encoding Mdm30 is missing. The labeling of Figure 3c and 3e are unclear/incomplete also the appropriate size of the bands are missing. The authors should indicate that they show that the DUB bind to Mdm30 f-box mutant NOT to Mdm30 WT (as implied here).

*Response R3.17: Data shown in Figure 3a, 3b, 3g and 4b have been obtained by employing a plasmid shuffling strategy. WT strains actually correspond to *mdm30Δ* strains covered by the MDM30 shuffling plasmid whereas *mdm30Δ* strains correspond to strains that have been cured from the MDM30 shuffling plasmid. These data thus already contain the control rescue by the plasmid encoding Mdm30. In the original manuscript, these aspects were notified in the figure legends and in the Methods section. In the revised manuscript, the main text has been edited to clearly state that we employed a plasmid shuffling strategy (Page 10, line 7) and the mention “MDM30 shuffling strains” has been added on Figures 3a, 3b, 3g and 4b. The new supplementary Figure 3a also demonstrates that the respiratory growth defect of *ubp2Δ* cells at 37°C is rescued by WT Ubp2 but not by the catalytic mutant C745S. Figure 3c and 3e have been reorganized and MW markers have been added. The issue with the co-IP experiment (Figure 3d) has been resolved as explained above (see response R2.7).*

13. Figure 4c: Was the quantification of number of zygotes only performed once? Error bars are missing and/or the number of cells analyzed.

Response R3.18: Out of 19 zygotes analyzed in this experiment, all were found to display total colocalization between mito-GFP and mito-RFP. Because no mitochondrial fusion defect was observed, this experiment was thus performed only once. In the revised manuscript, the assay has been repeated and the WT control strains have been included in the experiment. In all cases, 100% fusion efficiency was obtained which exempts any requirement for error bars.

14. Western blot of 4e is not convincing and does not fit to the quantification shown. Please improve and indicate size markers.

Response R3.19: Genomically tagged Ole1 only decreases by 25% in absence of Mdm30 which fits the slight decrease seen on the Western blot. In the original manuscript, this figure was likely misleading because the graph was beginning at 50%. This has now been corrected and this figure has been reorganized with size markers.

*The 25% decrease in Ole1 when Mdm30 is absent may be considered as marginal but is in fact quite significant. In terms of comparison, deletion of UBX2 or MGA2, two genes involved in the activation of the OLE1-pathway, induce decreases in expression of the endogenous Ole1 protein of 38 and 51%, respectively. These partial downregulations of Ole1 expression have drastic effects on lipids metabolism because of increased fatty acids saturation (Surma et al. 2013). While not as drastic, the effect of the 25% Ole1 decrease in *mdm30Δ* cells may be significant enough to modify lipids on outer membranes and interfere with their fusion. The main text has been edited to emphasize these points (Page 14, lines 3/4 and 8/9).*

15. Was the experiment in figure 5a only performed once? Error bars? 5b a quantification of the increase of the protein levels would be helpful. It would be interesting to know whether the Mdm30 expression levels increased or decreased also after UFA treatment in WT and *ubp2Δ* cells. 5b and 5d the protein sizes are missing. Significance of 5d (3 vs.4) should be tested.

*Response R3.20: The experiment in Figure 5a has now been repeated. As in Figure 4c, 100% fusion efficiency was obtained in all cases, which exempts any requirement for error bars. In Figure 5b, the amounts of Ubp2-HA and Fzo1 have been quantified and the extent of protein increase or decrease upon addition of Oleic Acid is now indicated together with size markers. The new supplementary Figure 5a shows that Mdm30 does not significantly vary after UFA treatment in WT and *ubp2Δ* cells. In Figure 5d, protein sizes have been added and statistical significance has been tested.*

16. The labelling for Figure 6a is absent (size?, channel?). Figure 6b one time experiment? Error bars? 6c, statistical analysis should be added for significance.

*Response R3.21: Labelling, channels and size bars have been added on Figure 6a. We apologize for the initial omission. Figure 6b is indeed a one-time experiment but fusion in the WT context is already shown to reach 100% in other figures (4c and 5a). The 2% partial fusion in this particular WT context (*mdm30Δ* shuffle strain covered by the MDM30 shuffle plasmid) is explained by probable loss of the MDM30 plasmid in one zygote out of 50 analyzed. The *mdm30Δ* context was otherwise assessed in three independent experiments in Figure 6c. Statistical significance is now tested in Figure 6c.*

17. Figure 7, a rescue of the growth defect with a plasmid expressing the protein to verify that the growth defect is specific should be added. Immunoblots with protein size markers are missing (see also remarks above for Figure 7df).

*Response R3.22: The new supplementary Figure 3a demonstrates that the respiratory growth defect of *ubp2Δ* cells at 37°C is rescued by WT Ubp2 but not by the catalytic mutant C745S (see response R3.17). Protein size markers have been included in Figure 7a and former Figure 7g (Figure 8d in the revised manuscript). Remarks on former Figures 7d and 7f have been considered (see response R3.4).*

18. Figure 7b: Why is OLE1 expression reduced in delta Ubp2? Please comment.

*Response R3.23: We thank the reviewer for raising this interesting point. Absence of Ubp2 results in loss of Rsp5 antagonism which would be expected to induce over-activation of the OLE1-pathway. Such over-activation is mimicked by addition of unsaturated fatty acids, which generates a feedback control that promotes down-regulation of Ole1 synthesis in response to increased desaturation (Hoppe et al. 2000 and Surma et al. 2013). This feedback loop may explain the decrease in OLE1 mRNAs in *ubp2Δ* cells. Alternatively, Rsp5 is known to auto-ubiquitylate itself in the absence of Ubp2 (Lam et al. 2013 and New supplementary Figure 4b), which may decrease its Ub ligase activity thus resulting in down-regulation of the OLE1-pathway.*

*While the exact reason for down-regulation of the OLE1-pathway in absence of Ubp2 remains to be fully understood, we have now analyzed expression of the Ole1 protein in WT and *ubp2Δ* cells (New supplementary Figure 8a). We found that Ole1 only decreases by 5 to 10% in the absence of Ubp2, which does not match the 60% decrease in mRNAs. This discrepancy is nonetheless easily explained by the fact that Mga2 has the ability to stabilize OLE1 mRNAs (Kandasamy et al. 2004). Absence of Ubp2 thus induces down-regulation of the OLE1-pathway but only slightly alters expression of the Ole1 protein. The main text has been edited to include these considerations (Page 18, line 17 ... Page 19, line 3).*

19. Suppl Fig. 6. Please add experiment without OA.

20. Suppl Fig. 7. Explain what is meant by X. Which dilutions of which strains.

*Response R3.25: In the original manuscript, the purpose of Supplementary Figure 7a with X strains was to provide proof that the last dilution in main Figure 7c (*ubp2Δ mga2Δ*) was performed on the same plate as control strains (WT, *ubp2Δ* and *mga2Δ*). In the revised manuscript, former Figure 7c has been removed and replaced by the full panel that was originally shown in former Supplementary Figure 7a. X has been replaced by the identity of the strains that represent two other *ubp2Δ mga2Δ* clones obtained from the mating between *ubp2Δ* and *mga2Δ* strains. Even though the respiratory growth of clone 1 is diminished as compared to that of clones 2 and 3, the new supplementary Figure 8b shows that, similar to clone 3 (Figure 8b), clone 1 is also subject to rescue by the FZO1 extra-copy.*

Figure for reviewers:

(A) Total protein extracts prepared from WT (MCY554), *ubp2Δ* (MCY1147), *mga2Δ* (MCY1078) and *ubp2Δ mga2Δ* (MCY1098) strains transformed with pRS414-TEF-FZO1-13MYC were analyzed by anti-Myc immunoblotting. MW in kDa are shown on the left of short and long exposures of immunoblots. In this experiment, levels of Fzo1-13Myc were not equalized in *UBP2* positive as compared to *UBP2* negative extracts. Detection of Fzo1-13Myc higher MW ubiquitylated species in

UBP2 negative extracts was consequently hampered. These species were nonetheless present in *ubp2Δ* and *ubp2Δ mga2Δ* cells (black arrows above 230 kDa).

(B) Ponceau staining of yeast extracts from isogenic *WT* (*RSP5*) and *rsp5Δ* strains. Total amounts of proteins are systematically lower in *rsp5Δ* extracts.

(C) [redacted]

(D) [redacted]

Reviewers' Comments:

Reviewer #1 (Remarks to the Author)

Figure 1: In the experiments related to the *rsp5* deletion mutant, kept alive by expressing the p90 fragment of *Spt23*, the conclusion is that (since *Fzo1* levels do not change in this mutant) *Rsp5* is not implicated in the regulation of *Fzo1* by *Ubp2*. The problem with this conclusion is that, as the authors are aware, by expressing the p90 fragment of *Spt23* you are constitutively leading to the production of presumably high levels of *OLE1* and unsaturated fatty acids, which may create a condition that precludes any conclusion about the role of *Rsp5* in directly ubiquitylating *Fzo1*. The authors further substantiate their conclusion by the fact that *Fzo1* has been shown to accumulate K48 ubiquitin chains, while *Rsp5* catalyzes primarily K63 chains. However, a recent paper provides an explanation for how *Rsp5* can lead to the formation of K48 chains (Nature Communications 2016, from T. Mayor's lab).

Figure 2C, 2D, Supplementary Figure 2D: I don't understand how any conclusions about *Ubp2* biochemical activities can be drawn when the *Ubp2* protein that was used for in vitro assays was derived by immunoprecipitation from yeast cells. It is not determined whether *Rsp5* or *Rup1* (an essential cofactor for *Rsp5-Ubp2* interaction) co-IP'd with *Ubp2* in these assays and whether these influenced the results of the deubiquitylation assays. This is a major issue and this greatly influences the already speculative conclusions on page 9 of the manuscript (Lines 5-21), which propose that *Ubp2* limits *Mdm30*-catalyzed extension of K48 chains on *Fzo1*.

Figure 3C. There are no error bars or indication of how many biological replicates were employed for this experiment. The effects here are two-fold with respect to levels of *Ubp2*, so this is critical.

It is not clear why the *Mdm30-Ubp2* interaction was not examined more directly, rather than only in co-IPs from cell lysates. The *Rsp5-Ubp2* interaction has been recapitulated in vitro, so why not confirm and further characterize the potential interaction with *Mdm30* in vitro? This would also allow experiments to be done to determine whether the interaction of *Ubp2* with *Rup1* (see below) and *Rsp5* is mutually exclusive with the *Mdm30* interaction.

The results shown in Figures 3e and f are marginal; the difference in half-life of *Ubp2* in the absence and presence of *Mdm30*, in particular. This leads to questioning the conclusion on page 12 lines 1-3: that accumulation of *Ubp2* via *mdm30* deletion leads to the respiratory defect.

Figure 4g and f: I don't understand why the same strains were not analyzed for both *OLE1* protein and for *OLE1* mRNA.

Page 14, lines 18 and 19: the sentence does not make sense and the meaning is unclear.

An important point not brought up by any of the reviewers in the first round of review is that if there really is a ménage à trois between *Mdm30* and *Ubp2* and *Rsp5*, then there is a predicted fourth partner, as well – *Rup1* – which is required for *Ubp2-Rsp5* interactions. I don't believe *Rup1* is never mentioned in the paper.

Reviewer #2 (Remarks to the Author)

I am in general satisfied with the author's responses to my points.

The have strengthened their story by providing better evidence of their claims.

-I have one remaining comments/questions concerning the in vitro deubiquitylation assays: no Ub2 K48 species are being generated during the assay, while Ub3 species are strongly increased. The fact that Ub3 are increased argues for Ubp2 being an endo-deubiquitylase. However, since no Ub2 species are generated, this looks more like an exo-ubiquitylase. Is there an explanation for that?

In other words: Ub5 can be broken down to Ub4+Ub1 (Fig. 2C), Ub6 and Ub7 can be broken down to Ub3 (plus something else presumably, Fig 2d), but Ub2 cannot be generated. This is puzzling and should be discussed.

-An extra effort can still be made on the writing to simplify many sentences.

Reviewer #3 (Remarks to the Author)

The authors have addressed all my concerns sufficiently. The manuscript has improved a lot by the changes and additional experiments and is now recommended for publication.

Minor points:

Figure 3d: MW marker is still missing and should be added

Responses to Reviewers' Comments

To accompany the point-by-point response to Reviewers' comments, below please find an overview of the main changes that have been incorporated into the study.

Overview of main changes

New data:

- Figure 1e: Effect of *RUP1* deletion on Fzo1 endogenous levels in response to R1.1, R1.2 and R1.8
- Figure 1f: Effect of *RUP1* deletion on Fzo1 ubiquitylation in response to R1.1, R1.2 and R1.8
- Figure 3c: Experiment repeated twice more and SD bars added (n=3) in response to R1.3 and R1.5
- Supplementary Figure 3d: Effect of *RUP1* deletion on Ubp2 endogenous levels in response to R1.4, R1.5 and R1.8
- Supplementary Figure 3e: Co-IP between Mdm30 and Ubp2 in *rup1Δ* cells in response to R1.4 and R1.8
- Figure A for reviewers: Ole1 protein levels in *rsp5Δ+spt23** cells in response to R1.1
- Figure B for reviewers: Spot assays with *rup1Δ* cells in response to R1.1, R1.2 and R1.8
- Figure C for reviewers: Co-IP between Fzo1 and Rsp5 in response to R1.1
- Figure D for reviewers: Expression of Mdm30-GST in *E.Coli* in response to R1.4

Modification of existing Figures:

- Figure 2c: Initial Figure 2e renamed as Figure 2c in response to R1.2
- Supplementary Figure 8a: Initial Figure 7b moved to supplementary Figure 8a in response to R2.2
- Figure 3d: MW markers have been added in response to R3.1

Text editing: The main text has been extensively edited in response to R2.2. Word files with track changes have been submitted together with the revised study. Changes introduced to comply with reviewers' suggestions and to comment on new data added in the manuscript are otherwise notified in the responses below.

We thank the reviewers for their assessment of our manuscript. We hope that with the new experiments and added clarifications in the revised study, our work will be viewed as suitable for publication in Nature Communications.

Point-by-point responses to reviewers' comments

Reviewer #1 (Remarks to the Author):

Figure 1: In the experiments related to the *rsp5* deletion mutant, kept alive by expressing the p90 fragment of Spt23, the conclusion is that (since Fzo1 levels do not change in this mutant) Rsp5 is not implicated in the regulation of Fzo1 by Ubp2. The problem with this conclusion is that, as the authors are aware, by expressing the p90 fragment of Spt23 you are constitutively leading to the production of presumably high levels of OLE1 and unsaturated fatty acids, which may create a condition that precludes any conclusion about the role of Rsp5 in directly ubiquitylating Fzo1. The authors further substantiate their conclusion by the fact that Fzo1 has been shown to accumulate K48 ubiquitin chains, while Rsp5 catalyzes primarily K63 chains. However, a recent paper provides an explanation for how Rsp5 can lead to the formation of K48 chains (Nature Communications 2016, from T. Mayor's lab).

Response R1.1: We understand the reviewer's concern about Ole1 levels in the RSP5 deletion mutant kept alive by expressing the p90 fragment of Spt23. In this mutant, however, the processing of Mga2 should not take place. The levels of Ole1 should therefore not be high but similar to those seen in mga2Δ cells (Surma et al. 2013) in which Fzo1 levels and mitochondrial fusion are not affected (Figure 7 in the present study). To test this, we analyzed the endogenous levels of genomically tagged Ole1 (Ole1-3FLAG) in Wild-type and rsp5Δ+Spt23 cells. This confirmed that, similar to mga2Δ cells, Ole1 levels remain lower in rsp5Δ+Spt23 cells than in WT cells (see Figure A for reviewer). With this clarification, we hope the reviewer agrees that all effects of UBP2 deletion on Fzo1 persist in rsp5Δ+Spt23 cells and are not altered by the absence of Rsp5.

Notably, the resulting conclusion that Rsp5 is not involved in the effects of UBP2 deletion on Fzo1 ubiquitylation and degradation is further supported by new data demonstrating that deletion of RUP1 does not mimic the effects of UBP2 deletion on Fzo1. Rup1 is established to promote the interaction between Ubp2 and Rsp5 and is thereby dedicated to regulate the reversal of Rsp5-mediated ubiquitylation specifically. Consistent with this, all Rsp5 substrates for which ubiquitylation was shown to be affected by UBP2 deletion have also systematically been shown to be affected by RUP1 deletion (Kee et al. 2005; Lam et al. 2013; Lam et al. 2009; Ren et al. 2007). Our new observation that the absence of Rup1 does neither impact Fzo1 levels (New Figure 1e) or ubiquitylation (New Figure 1f) nor recapitulates the respiratory growth defect of ubp2Δ cells at 37°C (see Figure B for reviewer), is thus in straightforward agreement with the notion that Rsp5 is not involved in the regulation of Fzo1 ubiquitylation and degradation by Ubp2.

In Figure 1, we strictly seek verifying whether the effects of UBP2 deletion on Fzo1 ubiquitylation and degradation may involve Rsp5 or not. We fully agree that our results do not allow excluding a role for Rsp5 in directly ubiquitylating Fzo1 in other circumstances and we do not claim or intend to demonstrate the contrary in the manuscript. In the course of our experiments, we nonetheless never managed observing any impact of Rsp5 on Fzo1 ubiquitylation and degradation or any binding between this E3 and the yeast mitofusin in co-immunoprecipitation assays (see Figure C for reviewer).

Please note that in the paper from T. Mayor and colleagues, the formation of K48 chains by Rsp5 is explicitly mentioned to take place upon heat shock at 45°C. Regardless of the temperature employed to perform our experiments in the manuscript, the patterns of Fzo1 ubiquitylation in WT and ubp2Δ cells were systematically

reproduced, including at 23°C (Figure 1C) where K48 activity of Rsp5 is predicted to be marginal. In the revised manuscript, the lines referring to the K63 specificity of Rsp5 have nonetheless been removed and replaced by a new paragraph focusing on the non-involvement of Rup1 (Page 7, lines 3-8).

Figure 2C, 2D, Supplementary Figure 2D: I don't understand how any conclusions about Ubp2 biochemical activities can be drawn when the Ubp2 protein that was used for in vitro assays was derived by immunoprecipitation from yeast cells. It is not determined whether Rsp5 or Rup1 (an essential cofactor for Rsp5-Ubp2 interaction) co-IP'd with Ubp2 in these assays and whether these influenced the results of the deubiquitylation assays. This is a major issue and this greatly influences the already speculative conclusions on page 9 of the manuscript (Lines 5-21), which propose that Ubp2 limits Mdm30-catalyzed extension of K48 chains on Fzo1.

Response R1.2: The reviewer is correct. According to previous studies, we cannot exclude that some Rup1, and to a lesser extent some Rsp5, were present in the Ubp2 preparations we employed for deubiquitylation of K63 and K48-chains in vitro. Recombinant Rup1 was previously shown to partially stimulate the ability of recombinant Ubp2 to disassemble ubiquitin chains in vitro (Kee et al. 2005). However, in Kee et al., the efficiency of the trimming reactions were clearly shown to be modulated by the amount of Ubp2 added to the reaction rather than by the presence or the absence of Rup1. In this context, any positive impact of Rup1 in our deubiquitylation reactions would only be partial and would not significantly contribute to the effects we observed, as opposed to the strong responses to redox variations we obtained in our assays and that match the intrinsic ubiquitin-protease activity of Ubp2. Assessing this potential contribution of Rup1 could be informative but its relevance to the present work is limited since our new data (Figure 1e, Figure 1f and Figure A for reviewer) indicate that Rup1 does not participate in the Ubp2-dependent regulation of Fzo1 ubiquitylation and degradation.

*The conclusion that Ubp2 limits Mdm30-dependent extension of K48 chains on Fzo1 does not result from speculation but is supported by Figure 2e (2c in the revised manuscript). This Figure shows that Mdm30 catalyzes formation of the Fzo1 ubiquitin doublet on Lysine 398 as the K398R mutation abolishes this ubiquitylation. The Fzo1 smear that migrates above the ubiquitin doublet in *ubp2Δ* cells shows the same features as the ubiquitin doublet: its detection is not only abolished in the absence of Mdm30 but also upon mutation of Lysine 398. This allows concluding that Ubp2 limits the extension of the Mdm30-dependent ubiquitin doublet on Lysine 398 of Fzo1. To emphasize this, figure 2e has been renamed as 2c and its corresponding description has been moved earlier in the text (Page 8, lines 5-11).*

Figure 3C. There are no error bars or indication of how many biological replicates were employed for this experiment. The effects here are two-fold with respect to levels of Ubp2, so this is critical.

Response R1.3: We agree that this is important. The experiment has been repeated twice more and SD bars (n=3) have been added (New Figure 3c). Our initial observation is confirmed.

It is not clear why the Mdm30-Ubp2 interaction was not examined more directly, rather than only in co-IPs from cell lysates. The Rsp5-Ubp2 interaction has been recapitulated in vitro, so why not confirm and further characterize the potential interaction with Mdm30 in vitro? This would also allow experiments to be done to

determine whether the interaction of Ubp2 with Rup1 (see below) and Rsp5 is mutually exclusive with the Mdm30 interaction.

Response R1.4: This is a good point. We built plasmids that allow for expression of Mdm30-GST in E.Coli. The recombinant protein was expressed following induction with IPTG. Unfortunately, it aggregated and could not be solubilized (see Figure D for reviewer). Similar results were obtained with a version of Mdm30-GST mutated in the F-box domain. In vitro binding assays could thus not be carried out.

To circumvent this issue, we generated new strains that lack the RUP1 gene to assess the endogenous level of genomically tagged Ubp2 in the absence of Rup1 and to determine whether the Mdm30-Ubp2 interaction would be compromised when the RUP1 gene is deleted. The new supplementary Figure 3d demonstrates that the level of endogenous Ubp2 increases in the absence of Mdm30 but remains similar to the level seen in WT cells upon deletion of RUP1. Rup1 is therefore not required for Ubp2 degradation. Consistent with this, the new supplementary Figure 3e indicates that the co-immunoprecipitation between Ubp2-HA and the F-box mutant of Mdm30-Myc is not affected in the absence of Rup1. The main text has been edited to report on these new observations (Page 11, line 22 – Page 12, line 2).

The results shown in Figures 3e and f are marginal; the difference in half-life of Ubp2 in the absence and presence of Mdm30, in particular. This leads to questioning the conclusion on page 12 lines 1-3: that accumulation of Ubp2 via mdm30 deletion leads to the respiratory defect.

Response R1.5: In the first round of review, the changes in Ubp2 levels and degradation between WT and mdm30Δ cells were already qualified as marginal. Here, we reiterate that while these changes are indeed low they clearly exist and are totally reproducible. They were seen in Figures 3c, 3d and 3e and are now also reproduced in new supplementary Figure 3d and the revised Figure 3c. In this context, qualifying these results as marginal sounds somewhat over-critic. In our view, these data do clearly establish that the level of Ubp2 is sensitively increased in the absence of Mdm30. The result shown in Figure 3f is totally consistent with these observations. The stabilization of Ubp2 in mdm30Δ cells is evident but it is the degradation of Ubp2 in WT cells that is weak. This explains the low increase of Ubp2 in the absence of Mdm30.

Importantly, these findings only suggest that accumulation of Ubp2 via MDM30 deletion could lead to the respiratory defect (Page 12; lines 5-6). The actual demonstration is provided by the fact that deletion of UBP2 or its over-expression, respectively decreases (Figures 3b and supplementary 3b) or enhances (Figures 3g and supplementary 3f) the respiratory growth defect seen in mdm30Δ cells. These results are those demonstrating that accumulation of Ubp2 participates in the respiratory defect of mdm30Δ cells. Moreover, they are further supported by the observation that Rsp5 overexpression rescues respiratory growth in cells lacking Mdm30 (Figure 4b), which phenocopies the effect seen with UBP2 deletion.

Figure 4b and f: I don't understand why the same strains were not analyzed for both OLE1 protein and for OLE1 mRNA.

Response R1.6: Regarding 4e, the corresponding mRNA levels were (and are still) shown in Supplemental 4c of the manuscript. This data clearly confirms the effect of Rsp5 overexpression on activation of the OLE1-pathway. As for 4f, the triple genomic insertion required to obtain the ubp2Δ mdm30Δ OLE1-MYC strain unfortunately did

not succeed after several attempts. We hope the reviewer understands that triple genomic insertions are rarely successful.

Page 14, lines 18 and 19: the sentence does not make sense and the meaning is unclear.

Response R1.7: We agree. This sentence has been deleted.

An important point not brought up by any of the reviewers in the first round of review is that if there really is a ménage à trois between Mdm30 and Ubp2 and Rps5, then there a predicted fourth partner, as well – Rup1 – which is required for Ubp2-Rsp5 interactions. I don't believe Rup1 is never mentioned in the paper.

Response R1.8: We thank the reviewer for having raised this point. As explained above, we have analyzed the involvement of Rup1 in both the regulation of Fzo1 ubiquitylation/degradation by Ubp2 (Figure 1e, Figure 1f and Figure A for reviewer) and the regulation of Ubp2 degradation by Mdm30 (new supplementary Figures 3d and 3e). In all cases, deletion of Rup1 had no observable effect indicating that Rup1 is neither implicated in the regulation of Fzo1 ubiquitylation/degradation nor in the regulation of Ubp2 degradation and the binding of the DUB to Mdm30.

Reviewer #2 (Remarks to the Author):

I am in general satisfied with the author's responses to my points.

The have strengthened their story by providing better evidence of their claims.

-I have one remaining comments/questions concerning the in vitro deubiquitylation assays:

no Ub2 K48 species are being generated during the assay, while Ub3 species are strongly increased. The fact that Ub3 are increased argues for Ubp2 being an endo-deubiquitylase. However, since no Ub2 species are generated, this looks more like an exo-ubiquitylase. Is there an explanation for that?

In other words: Ub5 can be broken down to Ub4+Ub1 (Fig. 2C), Ub6 and Ub7 can be broken down to Ub3 (plus something else presumably, Fig 2d), but Ub2 cannot be generated. This is puzzling and should be discussed.

Response R2.1: The reviewer is absolutely correct. The only rational explanation that accounts for this differential capacity of Ubp2 to cleave Ub4K48, Ub5K48 and Ub6 or Ub7k48 chains relates to the presumably differential capacity of the Ubp2 catalytic site to access a K48 isopeptide bond according to the length of the chain. In this context, Ubp2 likely binds Ub3K48 chains but becomes competent to cleave the isopeptide bond localized after the third ubiquitin if and only if a minimum of three additional ubiquitin moieties are present. If a single additional ubiquitin is present (Ub4K48 chains), the isopeptide bond between Ub3 and Ub4 cannot fit the catalytic site and no processing occurs. If two additional ubiquitin moieties are present (Ub5k48 chains), the Ub3-Ub4 linkage is still inaccessible but the Ub4-Ub5 bond can fit the catalytic site with low stability to generate Ub4K48 chains with low efficiency. If three additional ubiquitin moieties or more are present (Ub6K48 chains and longer), the Ub3-Ub4 linkage can then access the catalytic site with enhanced stability to generate Ub3K48 products with better efficiency. While likely, this explanation remains too speculative to be mentioned in the main text. However, the data with Ub5K48 chains clearly shows that Ub4 is generated with low efficiency because the expected decrease in Ub5 remains below detection. In contrast, the data with Ub(3-7)K48 chains shows that Ub3 is generated with higher efficiency because of the concomitant decrease in Ub7 and Ub6 that is detected. These features have been mentioned and emphasized in the main text (Page 9, lines 2-4 and lines 11-13).

-An extra effort can still be made on the writing to simplify many sentences.

Response R2.2: We agree. The main text has been extensively edited and many sentences have been moved or deleted for simplification (a word file with track changes has been submitted together with the revised manuscript). In the course of this editing, one data has been moved from the main to supplementary figures (Initial Figure 7b has now been included in supplementary Figure 8a). The text corresponding to the description of this figure has also been moved from the main text to the legend of supplementary figure 8a. We hope that with these changes, the manuscript reads now better and is suitable for publication.

Reviewer #3 (Remarks to the Author):

The authors have addressed all my concerns sufficiently. The manuscript has improved a lot by the changes and additional experiments and is now recommended for publication.

Minor points:

Figure 3d: MW marker is still missing and should be added

Response R3.1: The reviewer is correct. The figure has been edited with addition of MW markers.

Figure for reviewers:

(A) Total protein extracts from *OLE1-3FLAG* WT and *OLE1-3FLAG rsp5Δ+spt23** strains (2 clones each) were analyzed by anti-FLAG immunoblotting (upper panel). Equal loading of extracts was monitored by Ponceau staining (lower panel). MW in kDa are shown on the right of immunoblot and Ponceau panels. Note that the level of Ole1-3FLAG is significantly lower in *rsp5Δ+spt23** than in WT strains. This is consistent with the expected absence of Mga2 processing in *rsp5Δ+spt23** cells.

(B) Serial dilutions of *WT*, *ubp2Δ*, *rup1Δ* and *mdm30Δ* strains (BY 4741 background) in the presence of glucose or glycerol as the sole carbon source at 23, 30 and 37°C. Note that deletion of RUP1 does not phenocopy the respiratory growth defect of *ubp2Δ* cells at 37°C.

(C) Co-IP between Fzo1-HA and Rsp5 fails to be detected. *WT* and *FZO1-HA* cells were lysed and lysates were subjected to co-IP with anti-HA antibody followed by immunoblotting with anti-HA or anti-Rsp5as indicated. Left panels, lysates (10% input of IP); right panels, immunoprecipitates. MW in kDa are shown on the right of immunoblots. Note that the level of Rsp5 in immunoprecipitates is similar whether Fzo1-HA is present or not.

(D) Expression of Mdm30-GST in B121 *E. Coli* cells. Crude extracts were prepared from cells transformed with an MDM30-GST expression plasmid and grown either in the presence (Induced) or in the absence (Non induced) of IPTG. The extract prepared from induced cells was further centrifuged at 10 000g yielding a Supernatant (Sup) and a Pellet fraction that contain the soluble and aggregated recombinant proteins, respectively. Proteins of all samples were separated by SDS-PAGE 8% and labelled by Coomassie staining. MW in kDa are shown on the left of the gel. Note that Mdm30-GST was well expressed but that all the protein was recovered in the pellet fraction.

Reviewers' Comments:

Reviewer #1:

Remarks to the Author:

The authors have thoroughly addressed the comments of all of the reviewers and I hope the authors will agree that process has resulted in a much improved manuscript.

Reviewer #2:

Remarks to the Author:

I am entirely satisfied by the revision.

A special effort has been made for the clarity of the text, which is to be commended.

Congratulation to the authors for their nice story.

REVIEWERS' COMMENTS:

Reviewer #1 (Remarks to the Author):

The authors have thoroughly addressed the comments of all of the reviewers and I hope the authors will agree that process has resulted in a much improved manuscript.

Response: We do agree that the revisions led to a much improved manuscript. We thus thank the reviewers for evaluating our study and for their suggestions.

Reviewer #2 (Remarks to the Author):

I am entirely satisfied by the revision.

A special effort has been made for the clarity of the text, which is to be commended.

Congratulation to the authors for their nice story.

Response: We would like to thank Reviewer 2 for his very constructive comments.